# MCARE enhances SERCA1 activity in fast-twitch muscle to maintain calcium handling and muscle integrity

Takashi Sasaki [1,2] ✉, Hayataka Takase[2], Michinori Koebis[3], Atsu Aiba [3], Yu Takahashi[1], Yoshio Yamauchi [1,2] & Ryuichiro Sato[1,2] ✉

The release of $Ca^{2+}$ from the sarcoplasmic reticulum into the cytoplasm, followed by its reuptake by sarco/endoplasmic reticulum $Ca^{2+}$ ATPase (SERCA), is critical for the muscle contraction-relaxation cycle. In this study, we identify a small transmembrane protein, predominantly expressed in fast-twitch muscles, which regulates SERCA1 activity. This protein, termed muscle-enriched $Ca^{2+}$ regulator (MCARE), enhances SERCA1 function by competitively inhibiting myoregulin, a muscle-specific micropeptide that otherwise suppresses SERCA1 activity. By facilitating more efficient $Ca^{2+}$ clearance from the cytoplasm, MCARE accelerates muscle relaxation. *Mcare*-deficient mice exhibit symptoms resembling muscular dystrophy, including progressive muscle wasting in fast-twitch muscles, reduced muscle strength, and increased susceptibility to exercise-induced muscle damage. Notably, these mice also present with distinctive rippling muscle contractions. Our findings establish MCARE as a key regulator of SERCA1 activity, essential for maintaining $Ca^{2+}$ homeostasis and the functional integrity of fast-twitch muscle fibers.

Intracellular $Ca^{2+}$ is pivotal in various aspects of cell physiology, including neuronal transmission, apoptosis, gene transcription, and cell migration[1,2]. The essential role of $Ca^{2+}$ is underscored by the efforts of cells to maintain lower cytosolic $Ca^{2+}$ levels, generally under 100 nM in the resting state. In skeletal muscle, contraction and relaxation are regulated by $Ca^{2+}$. The process begins with an action potential generated by motor neurons, which propagates across the muscle cell membrane and activates voltage-gated calcium channels, allowing $Ca^{2+}$ influx into the muscle cells. This influx activates the ryanodine receptor (RyR), prompting the release of stored $Ca^{2+}$ from the sarcoplasmic reticulum (SR) into the cytoplasm. The released $Ca^{2+}$ subsequently binds to troponin, initiating the sliding of myosin and actin filaments, leading to muscle contraction. As the action potential subsides, sarcoplasmic/endoplasmic reticulum calcium ATPase (SERCA) actively transports $Ca^{2+}$ back into the SR, enabling muscle relaxation.

SERCA is the major transmembrane ion transporter responsible for $Ca^{2+}$ reuptake from the cytosol into the ER/SR. In vertebrates, SERCA is encoded by three gene families, SERCA1, 2, and 3, each with a highly conserved primary structure but distinct tissue expression patterns[3]. SERCA1 and SERCA2 are predominantly found in muscle tissue, with SERCA1 mainly expressed in fast-twitch skeletal muscles and SERCA2 in slow-twitch skeletal and cardiac muscles. In contrast, SERCA3 is expressed in various tissues, such as endothelial and epithelial cells and platelets, but not in skeletal muscle cells. SERCA pump activity is regulated by the direct interaction of small single transmembrane micropeptides that overlap in their expression distribution with their respective isoforms. Myoregulin (MRLN), a micropeptide expressed mainly in fast-twitch muscle, inhibits SERCA1 activity, and *MRLN* KO mice show improved exercise performance and $Ca^{2+}$ handling in the skeletal muscle[4]. The micropeptides sarcolipin and

[1]Food Biochemistry Laboratory, Department of Applied Biological Chemistry, Graduate School of Agricultural and Life Sciences, The University of Tokyo, Tokyo, Japan. [2]Nutri-Life Science Laboratory, Department of Applied Biological Chemistry, Graduate School of Agricultural and Life Sciences, The University of Tokyo, Tokyo, Japan. [3]Laboratory of Animal Resources, Center for Disease Biology and Integrative Medicine, Graduate School of Medicine, The University of Tokyo, Tokyo, Japan. ✉e-mail: atsasaki@g.ecc.u-tokyo.ac.jp; roysato@g.ecc.u-tokyo.ac.jp

phospholamban inhibit SERCA2 activity in slow-twitch skeletal muscle and cardiac muscle, respectively[5], whereas the dwarf open reading frame (DWORF) enhances SERCA2-dependent $Ca^{2+}$ reuptake by displacing these inhibitory micropeptides and directly increasing SERCA's enzymatic activity[6–8]. The role of DWORF in cardiomyocytes has been well studied, with a previous report demonstrating that its overexpression in a mouse model of dilated cardiomyopathy enhances $Ca^{2+}$ cycling and myocyte contractility, thereby preventing the development of heart failure[9]. Therefore, SERCA and its activity-modulating micropeptides play indispensable roles as key regulators of muscle function, intricately controlling both contraction and relaxation.

In this study, we identified transmembrane protein 233 (Tmem233), a small protein predominantly expressed in fast-twitch muscle, as a regulator of skeletal muscle calcium signaling, which we termed muscle-enriched $Ca^{2+}$ regulator (MCARE). MCARE binds to SERCA1 on the SR, displacing MRLN and enhancing the enzymatic activity of SERCA1, thereby accelerating muscle relaxation. *Mcare* KO mice exhibited progressive muscle atrophy, reduced muscle strength, elevated serum creatine kinase (CK) levels, rippling muscle contractions, and exacerbated exercise-induced muscle damage. These data demonstrate that MCARE enhances SERCA1 activity, crucial for maintaining the functional integrity of fast-twitch skeletal muscle fibers.

## Results

### MCARE is a fast-twitch muscle-specific transmembrane protein
*Mcare* is an evolutionarily conserved gene across vertebrates and belongs to a dispanin family characterized by two transmembrane domains (Fig. 1a)[10]. However, its physiological functions remain unknown.

To determine the tissue distribution of *Mcare*, we measured its mRNA expression in mice. The gene expression of *Mcare* is highly abundant in the gastrocnemius muscle, predominantly composed of fast-twitch fibers, while exhibiting comparatively lower levels in the soleus muscle, which comprises ~40% of slow-type fibers (Fig. 1b). Notably, the expression pattern of *Mcare* closely paralleled that of *SERCA1* and its inhibitory micropeptide *MRLN*. In addition, *Mcare* was detected in whole skeletal muscle tissue, but was absent in mononuclear cell populations isolated from muscle, such as satellite cells and macrophages (Supplementary Fig. 1a). This muscle-specific expression pattern of *Mcare* is conserved across species, including humans and *Xenopus*, highlighting its potential physiological importance (Supplementary Fig. 1b, c)[11,12]. Consistent with the mRNA distribution, fast-twitch muscles such as the gastrocnemius, quadriceps, extensor digitorum longus (EDL), and tibialis anterior (TA) exhibited substantial MCARE protein levels compared with other tissues (Fig. 1c and Supplementary Fig. 1d). In contrast, MCARE was undetectable in C2C12 myoblasts and myotubes, as also shown in Fig. 1c.

### MCARE and SERCA1 share the same subcellular localization
C2C12 myoblasts were transfected with EGFP-MCARE to examine the subcellular distribution of MCARE. Its localization overlapped with co-expressed mCherry-SERCA1a at the SR and perinuclear regions, but not with the mitochondria or lysosomes (Fig. 1d and Supplementary Fig. 2a). Next, we investigated MCARE distribution in muscle fibers by electroporating the EGFP-MCARE expression vector and mCherry-SERCA1a expression vector into mouse TA muscle. The resulting fluorescence signals revealed that EGFP-MCARE and mCherry-SERCA1a co-localized, forming distinct transverse and longitudinal striations characteristic of the SR architecture (Fig. 1e). To further corroborate the subcellular localization of endogenous MCARE, we isolated microsomal fractions from skeletal muscles. In whole muscle lysates, markers for various organelles, including Caveolin-3, N-cadherin, Lamin A/C, Tom20, GM130, and HSP90, were detected. In contrast, the microsomal fractions exhibited strong enrichment of SERCA1, and MCARE was also detected (Fig. 1f). These findings reinforce that

endogenous MCARE predominantly localizes to the SR in skeletal muscle.

The intracellular localization of the N-terminal region of MCARE was determined using a differential permeabilization method. C2C12 myoblasts expressing Myc-SERCA1a and Flag-MCARE were subjected to selective or complete permeabilization using digitonin or Triton X-100, respectively. Subsequently, the cells were labeled with anti-Myc and anti-Flag antibodies. The N-terminal Myc-tag of SERCA1a was detected by the anti-Myc antibody following permeabilization with either digitonin or Triton X-100, whereas the N-terminal Flag-tag of MCARE was detectable only after Triton X-100 treatment (Fig. 1g). These results indicate that the N-terminus of MCARE protrudes into the SR lumen in an orientation opposite to the N-terminus of SERCA1a. A similar experiment was performed for the C-terminus of MCARE; however, the intracellular localization of its C-terminus could not be determined, possibly due to destabilization caused by the addition of epitope tags (Supplementary Fig. 2b, c). Nevertheless, based on these findings and the structural characteristics of the dispanin protein family, it is hypothesized that both terminals of MCARE are oriented toward the SR lumen (Supplementary Fig. 2d).

### MCARE inhibits intracellular $Ca^{2+}$ upregulation by EPS and promotes muscle relaxation
The SR is an intracellular organelle responsible for the regulation of cytosolic $Ca^{2+}$ concentration, which is essential for muscle contraction. To elucidate the function of MCARE in the SR membrane, its effect on $Ca^{2+}$ flux was assessed by monitoring intracellular $Ca^{2+}$ dynamics during electrical pulse stimulation (EPS). C2C12 myotubes loaded with the $Ca^{2+}$ indicator Fluo-8 were used for this analysis. In both control and MCARE-expressing C2C12 myotubes, EPS (23 V, 2 ms) elicited a transient increase in $Ca^{2+}$ fluorescence with no significant difference in the proportion of responsive myotubes between the two groups (Fig. 2a–c, and Supplementary Movie 1). However, the fluorescence changes (Max $\Delta F/F_0$) were significantly diminished in the presence of MCARE, indicating its involvement in $Ca^{2+}$ cycling between the SR and the cytosol (Fig. 2d). To further evaluate the effect of MCARE under stronger stimulation conditions, we applied a high-intensity EPS protocol (23 V, 6 ms), in addition to the standard EPS condition (23 V, 2 ms) used throughout this study. Under this stronger stimulus, both control and MCARE-expressing myotubes exhibited increased $Ca^{2+}$ fluorescence responses compared with the standard EPS (Supplementary Fig. 3a–c). Although the difference in Max $\Delta F/F_0$ between the two groups became relatively smaller, the response remained significantly lower in the MCARE group. To assess the impact of MCARE on the decay phase of the $Ca^{2+}$ transient, we calculated the time constant (tau) of fluorescence decay following high-intensity EPS. This analysis revealed that tau was significantly reduced in MCARE-expressing myotubes, indicating that MCARE accelerates cytosolic $Ca^{2+}$ clearance after EPS (Fig. 2e). As MCARE did not affect cell morphology, differentiation index, or fusion index, it is unlikely that these effects were due to impaired muscle maturation (Supplementary Fig. 4a). Furthermore, we confirmed that MCARE did not alter the expression of RyR, SERCA1, and SERCA2 (Supplementary Fig. 4b).

Next, motion analysis was used to measure contraction displacement as well as contraction and relaxation velocities of MCARE-expressing C2C12 myotubes in response to EPS. These parameters were calculated by tracking the movement of a selected point on myotubes over time (Fig. 2f). Consistent with our finding that MCARE attenuated the $Ca^{2+}$ response to EPS, C2C12 myotubes expressing MCARE exhibited a notable decrease in muscle contraction displacement following EPS (Fig. 2g and Supplementary Movie 2). Furthermore, the maximum velocities of muscle contraction and relaxation were significantly reduced in the presence of MCARE (Fig. 2h). Because both contraction and relaxation velocities were strongly correlated with contraction displacement, the data were reanalyzed using

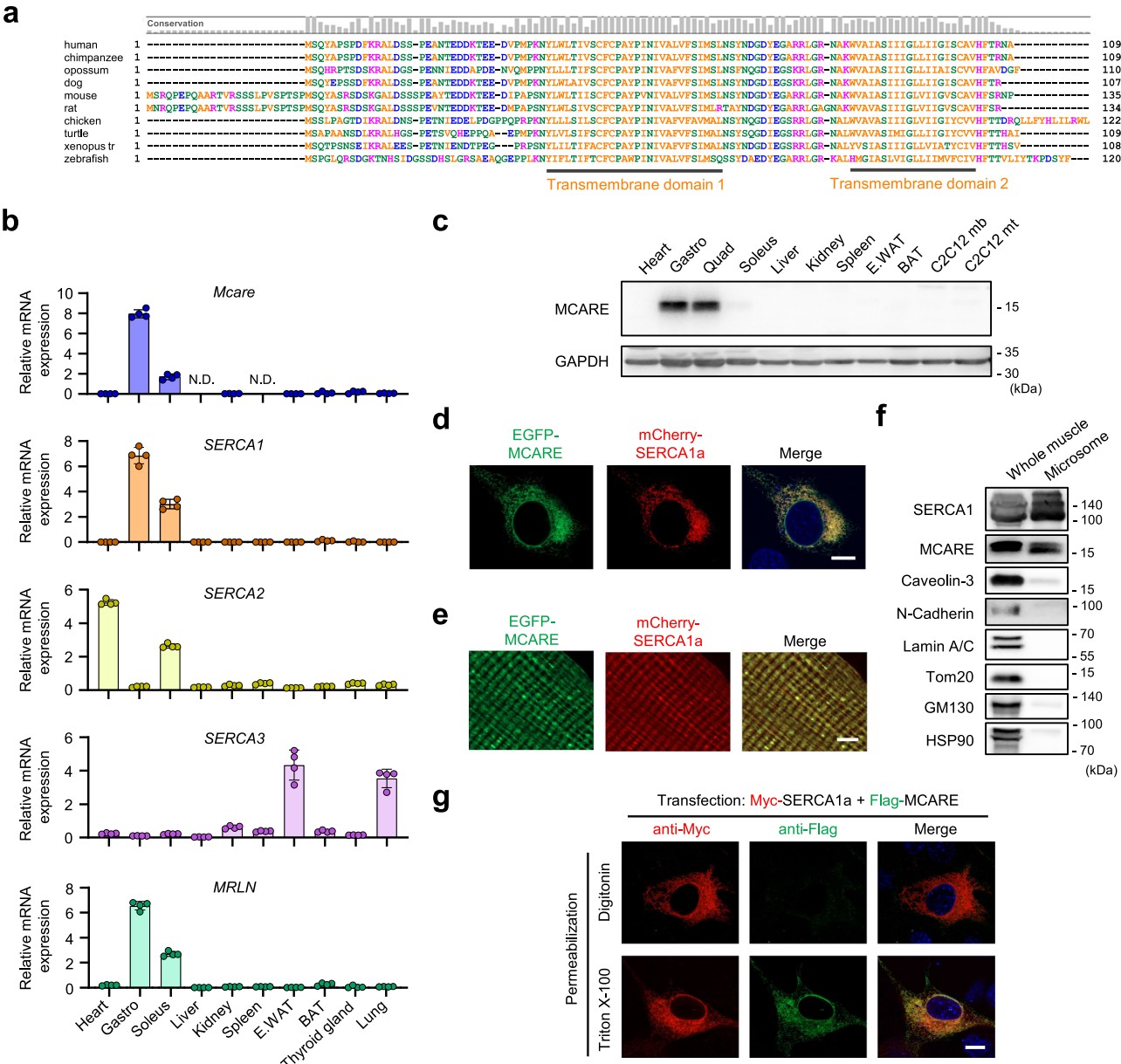

**Fig. 1 | MCARE is predominantly expressed in fast-twitch muscle and localizes to the SR. a** Alignment of the amino acid sequence of MCARE from different species. The conservation rate of each amino acid is indicated at the top. **b** qRT-PCR showing relative mRNA expression of *Mcare*, *SERCA* isoforms, and *MRLN* across adult mouse tissues. N.D. not detected. n = 4 mice. Error bars represent mean ± SD. **c** Western blot of adult mouse tissues and C2C12 myoblasts (mb) and myotubes (mt) with MCARE- and GAPDH-specific antibodies. Each lane contains pooled lysates from three independent biological sources (tissues or wells). **d** Fluorescence microscopy images of C2C12 myoblasts expressing EGFP-tagged MCARE and mCherry-tagged SERCA1a. Scale bar, 10 μm. **e** Fluorescence microscopy images of isolated mouse TA muscle fibers after in vivo electroporation with EGFP-MCARE and mCherry-SERCA1a. Scale bar, 5 μm. **f** Western blot of whole skeletal muscle tissue and its corresponding microsomal fraction from a mouse, probed with the indicated antibodies. **g** Myc-SERCA1a and Flag-MCARE were transiently expressed in C2C12 myoblasts. Proteins were detected by immunofluorescence analysis in completely permeabilized (Triton X-100) and selectively permeabilized (digitonin) cells. Scale bar, 10 μm. Gastro gastrocnemius, Quad quadriceps, E. WAT epididymal white adipose tissue, BAT brown adipose tissue.

analysis of covariance (ANCOVA) with contraction displacement as a covariate. Although contraction velocity did not differ between the groups, the regression slope for relaxation velocity was significantly steeper in the MCARE-expressing group (Fig. 2i). The corrected analysis, adjusted for contraction displacement, led to the same conclusion that MCARE affects relaxation velocity but not contraction velocity (Fig. 2j). To further characterize the effect of MCARE on muscle relaxation dynamics, we analyzed the time-dependent changes in displacement following peak contraction. This analysis allowed us to determine the half-relaxation time ($T_{50}$), defined as the time required for displacement to decline to 50% of the peak value after EPS. Consistent with the increased relaxation velocity observed in MCARE-expressing myotubes, $T_{50}$ was significantly shorter in the MCARE group (Fig. 2k), indicating that MCARE accelerates the relaxation phase following contraction.

Skeletal muscle contraction is initiated by a rapid increase in the cytosolic $Ca^{2+}$ concentration, whereas relaxation is achieved by the uptake of cytosolic $Ca^{2+}$ into the SR by SERCA[13]. To evaluate the impact of SERCA inhibition on $Ca^{2+}$ dynamics, we performed calcium imaging following thapsigargin treatment. Thapsigargin significantly increased the peak cytosolic $Ca^{2+}$ level, indicating impaired reuptake of $Ca^{2+}$ into the SR (Supplementary Fig. 5a, b). We next assessed the effects of thapsigargin on contraction and relaxation dynamics using motion analysis. Contraction displacement was comparable between control

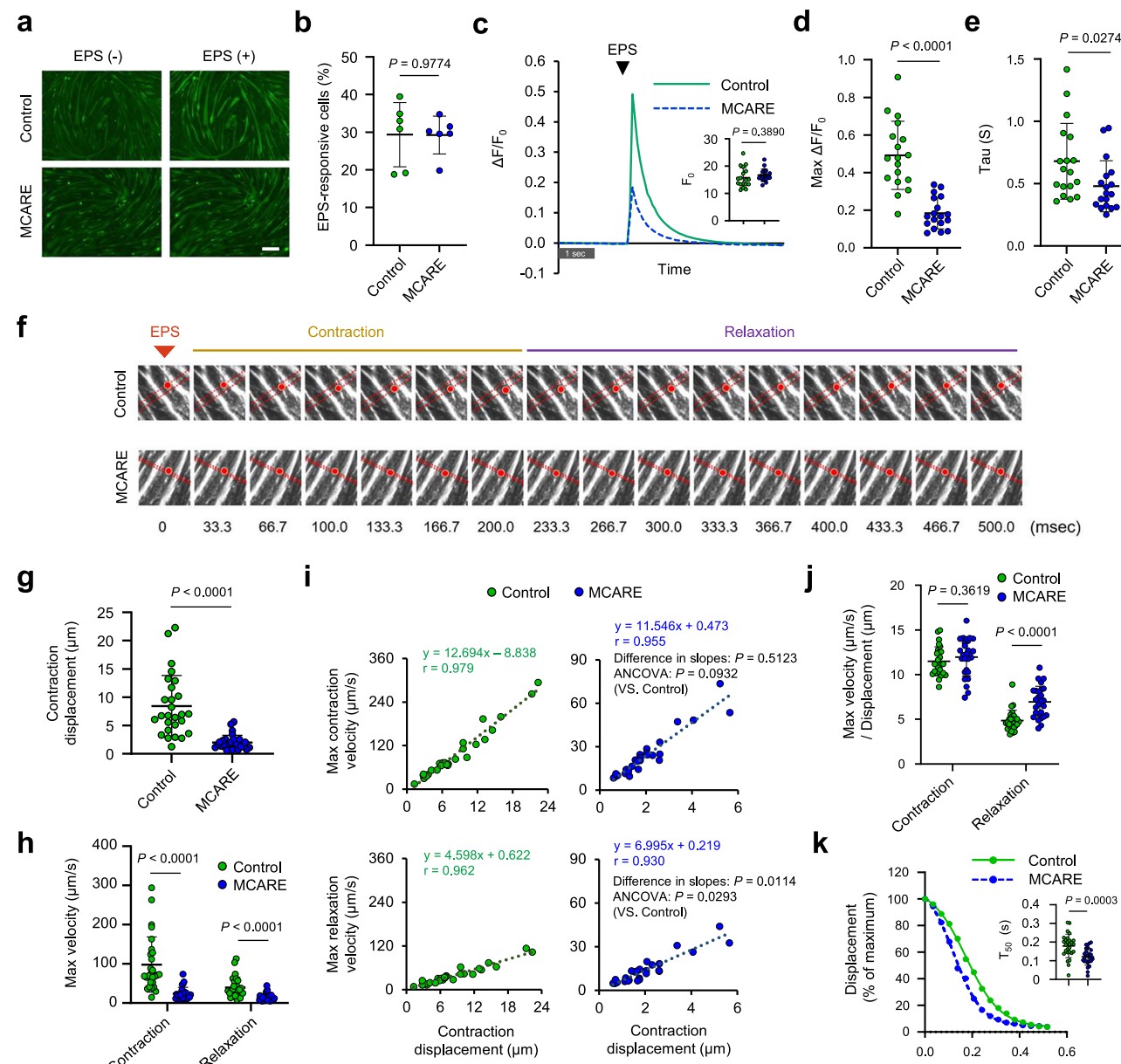

**Fig. 2 | MCARE attenuates EPS-induced intracellular Ca²⁺ elevation and promotes muscle relaxation. a** Fluo-8 fluorescence images recorded during EPS (23 V, 2 ms) in C2C12 myotubes expressing LacZ (control) or MCARE. Scale bar, 200 μm. **b** Quantification of the proportion of EPS-responsive myotubes in control and MCARE-expressing groups. The proportion of responsive cells was calculated for each field of view following EPS. $n = 6$ fields per group. **c** Changes in Ca²⁺ fluorescence intensity ($\Delta F/F_0$) in response to EPS. Data include $\Delta F/F_0$ traces and baseline fluorescence ($F_0$) from $n = 18$ myotubes analyzed from six fields of view. **d** Maximum amplitude (Max $\Delta F/F_0$) of the EPS-induced Ca²⁺ fluorescence, analyzed from the same dataset as in (**c**). $n = 18$ myotubes. **e** Time constant of Ca²⁺ signal decay was determined by single-exponential fitting of fluorescence traces after high-intensity EPS (23 V, 6 ms). $n = 18$ myotubes analyzed from six fields of view. **f** Example of tracking selected points (red dots) on a myotube. The range of movement is represented by two dotted lines, and this distance is defined as the contraction displacement. **g** Contraction displacement of the control and MCARE-expressing C2C12 myotubes in response to EPS (23 V, 2 ms). **h** Maximum contraction and relaxation velocities. **i** Correlation analysis of contraction displacement with maximum contraction and relaxation velocities. Differences in regression slopes between control and MCARE groups were evaluated using two-tailed ANCOVA with contraction displacement as a covariate (contraction × displacement interaction: $F(1, 51) = 0.44$, $P = 0.5123$; relaxation × displacement interaction: $F(1, 51) = 6.91$, $P = 0.0114$). The overall ANCOVA results for group effects are shown (contraction: $F(1, 51) = 2.93$, $P = 0.0932$; relaxation: $F(1, 51) = 5.03$, $P = 0.0293$). **j** Normalized contraction and relaxation velocities, adjusted by contraction displacement. **k** Time-dependent changes in displacement from the peak contraction and $T_{50}$ after EPS. All analyses in (**g**–**k**) were performed using the same dataset comprising $n = 27$ myotubes analyzed from nine fields of view. Error bars represent mean ± SD. Statistical significance was assessed using a two-tailed unpaired Student's *t*-test (**b**, **c**, **d**, **e**, **g**, **h**, **j**, and **k**) or two-tailed ANCOVA (**i**).

and thapsigargin-treated groups. Maximum contraction velocity was slightly reduced (11.7% decrease), whereas relaxation velocity was markedly decreased (48.7% decrease) by thapsigargin treatment (Supplementary Fig. 5c, d). These findings were confirmed by ANCOVA using contraction displacement as a covariate, as well as by normalization of velocity to displacement, both of which demonstrated that

relaxation velocity was markedly impaired, while the effect on contraction velocity was minimal (Supplementary Fig. 5e, f). Furthermore, $T_{50}$ was significantly prolonged in the thapsigargin-treated group, supporting delayed muscle relaxation (Supplementary Fig. 5g). Collectively, these findings indicate that changes in relaxation velocity and dynamics in this experimental setting are closely associated with

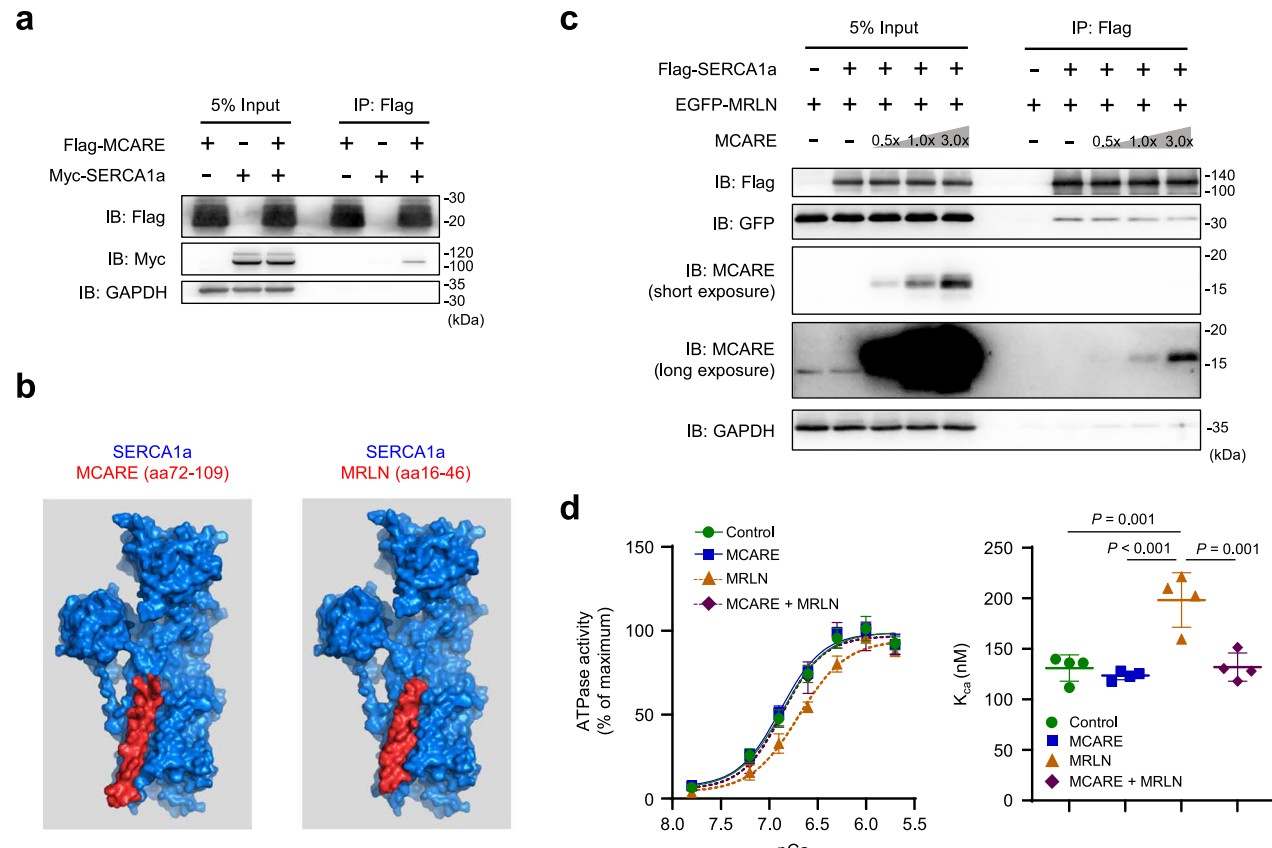

**Fig. 3 | MCARE displaces MRLN and enhances SERCA activity. a** Western blot of 5% input and immunoprecipitated (IP) Flag fractions from HEK293 cells transfected with Flag-MCARE and Myc-SERCA1a. **b** Protein docking simulation using ClusPro (https://cluspro.org) predicted that MCARE and MRLN occupy the same groove in SERCA1a. **c** Western blot of 5% input lysates and IP Flag fractions from HEK293 cells transfected with Flag-SERCA1a, EGFP-MRLN, and MCARE. **d** SERCA activity and affinity for $Ca^{2+}$ ($K_{ca}$) were measured in membrane vesicles from HEK293 cells co-transfected with SERCA1a and the indicated constructs. n = 4 biological replicates. Error bars represent mean ± SD. Statistical significance was assessed using two-way ANOVA with Tukey's multiple comparisons test.

SERCA activity and raise the possibility that MCARE plays a role in enhancing SERCA-dependent muscle relaxation.

## MCARE associates with SERCA1a and competitively inhibits MRLN

Given that MCARE did not alter SERCA protein expression levels in C2C12 myotubes, we hypothesized that MCARE directly binds to SERCA on the SR membrane to regulate its pump activity. To examine the interaction between these proteins, we performed co-immunoprecipitation experiments using HEK293 cells co-expressing Flag-MCARE and Myc-SERCA1a, which revealed a stable interaction between MCARE and SERCA1a (Fig. 3a). In contrast, a clear interaction between MCARE and SERCA2 was not detected (Supplementary Fig. 6). Automated protein docking simulation of transmembrane domain 2 of MCARE with the crystal structure of SERCA1 suggested that MCARE could bind to the groove formed within the SR membrane by SERCA1 (Fig. 3b). Notably, this groove is also required for the binding of MRLN to SERCA1, which inhibits $Ca^{2+}$ uptake[4]. Next, we examined whether MCARE expression influences the complex formation between SERCA1 and MRLN. Our results demonstrated that MCARE co-expression inhibited the binding of Flag-SERCA1a to EGFP-MRLN in a dose-dependent manner, suggesting that MCARE and MRLN compete for binding to SERCA1a (Fig. 3c). To determine whether MCARE attenuates the effect of MRLN on SERCA activity, we measured $Ca^{2+}$-ATPase activity by using SERCA1a-expressing HEK293 cells. While MCARE expression alone did not affect SERCA activity, it suppressed the MRLN-mediated inhibition of SERCA and significantly increased pump activity in the presence of MRLN (Fig. 3d).

## *Mcare* deficiency impairs $Ca^{2+}$ handling and induces muscle dysfunction in vivo

To investigate the role of MCARE in vivo, we generated *Mcare* knockout (KO) mice using the CRISPR/Cas9 system (Supplementary Fig. 7a–c). The loss of MCARE did not affect the protein levels of SERCA1 or RyR in skeletal muscle, consistent with our finding that overexpression of MCARE in C2C12 myotubes did not alter the expression of these proteins (Supplementary Fig. 7b). In the context of our observations that MCARE suppressed $Ca^{2+}$ upregulation by EPS in C2C12 myotubes, we evaluated whether the loss of MCARE affected EPS-induced $Ca^{2+}$ elevation in primary myotubes from these mice. As expected, calcium imaging data showed that *Mcare* deficiency promoted an increase in cytosolic $Ca^{2+}$ concentration by EPS (Fig. 4a–d). To further assess the functional role of endogenous MCARE, we performed motion analysis of these primary myotubes. Although there were no significant differences in contraction displacement, maximum contraction velocity, or maximum relaxation velocity between the two groups, ANCOVA using contraction displacement as a covariate revealed that the regression slope for relaxation velocity was significantly reduced in KO-derived myotubes (Fig. 4e–g). Consistent with this, relaxation velocity normalized to contraction displacement was significantly lower in the KO group (Fig. 4h). Moreover, $T_{50}$ was significantly prolonged in KO myotubes compared with WT controls (Fig. 4i). These findings indicate that endogenous MCARE also contributes to the regulation of SERCA-mediated muscle relaxation dynamics.

Given that our data (Fig. 4a–i) suggest reduced SERCA activity in the skeletal muscle of *Mcare* KO mice, we next investigated whether impaired $Ca^{2+}$ reuptake contributes to muscle pathology in vivo.

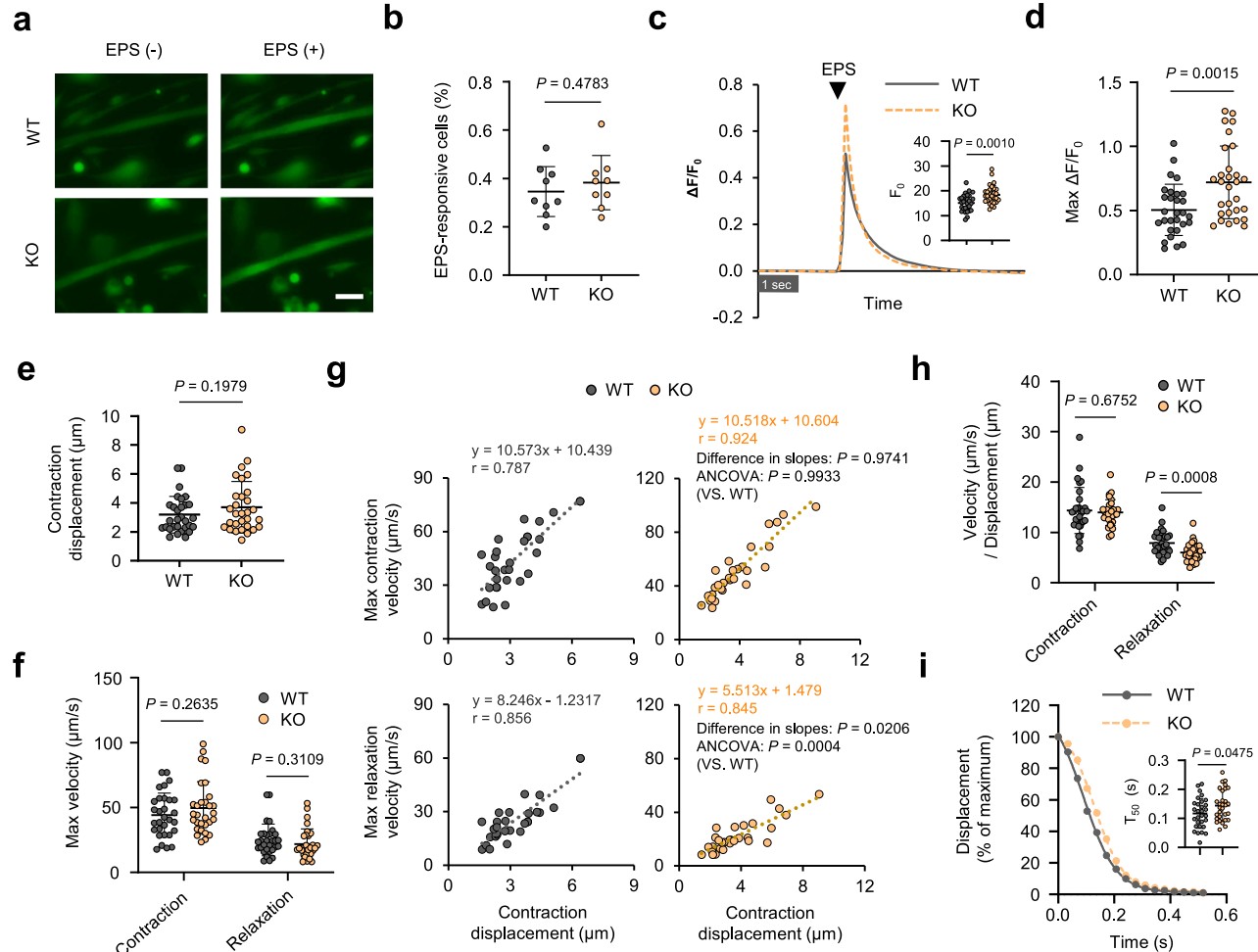

**Fig. 4 | Loss of *Mcare* impairs EPS-evoked Ca²⁺ responses and muscle relaxation in primary myotubes. a** Fluo-8 fluorescence images recorded during EPS in primary myotubes from WT and KO mice. Scale bar, 50 μm. **b** Quantification of the proportion of EPS-responsive primary myotubes from WT and KO mice. The proportion of responsive myotubes was calculated for each field of view (9 fields per group) following EPS. **c** Changes in Ca²⁺ fluorescence intensity in response to EPS. Data include $\Delta F/F_0$ traces and $F_0$ from n = 29 myotubes analyzed from ten fields of view. **d** Maximum $\Delta F/F_0$ of EPS-induced Ca²⁺ fluorescence, analyzed from the same dataset as in (**c**). n = 29 myotubes. **e** Contraction displacement of primary myotubes from WT and KO mice. **f** Maximum contraction and relaxation velocities. **g** Correlation analysis of contraction displacement with maximum contraction and

relaxation velocities. Differences in regression slopes between WT and KO groups were tested using two-tailed ANCOVA with contraction displacement as a covariate (contraction × displacement interaction: $F(1, 57) < 0.01$, $P = 0.9741$; relaxation × displacement interaction: $F(1, 57) = 5.67$, $P = 0.0206$). The overall ANCOVA results for group effects are shown (contraction: $F(1, 57) < 0.01$, $P = 0.9933$; relaxation: $F(1, 57) = 13.94$, $P = 0.0004$). **h** Normalized contraction and relaxation velocities, adjusted by contraction displacement. **i** Time-dependent changes in displacement from the peak contraction and $T_{50}$ after EPS. All analyses in (**e–i**) were performed using the same dataset comprising n = 30 myotubes analyzed from ten fields of view. Error bars represent mean ± SD. Statistical significance was assessed using two-tailed unpaired Student's *t*-test (**b**, **c**, **d**, **e**, **f**, **h**, and **i**) or two-tailed ANCOVA (**g**).

Notably, mutations in SERCA1 cause Brody disease, a human myopathy characterized not only by delayed muscle relaxation but also by reduced muscle mass and strength, along with elevated serum CK levels[14,15]. These clinical features suggest that impaired SERCA1 function can lead to complex muscle dysfunction beyond simple contractile defects. *Mcare*-deficient mice showed similar food intake, energy expenditure, and RER compared with WT mice (Supplementary Fig. 8a, b), and there were no significant differences in body weight, liver weight, or white adipose tissue weight between the two groups (Supplementary Fig. 9). Despite these systemic similarities, KO mice exhibited progressive muscle atrophy that was most pronounced in proximal muscles, such as the quadriceps, whereas the distal TA muscle—although expressing high levels of MCARE—did not show significant atrophy (Fig. 5a). Notably, significant reductions in quadriceps weight were evident as early as three months of age. The absence of classical atrophy-related gene upregulation suggests that this phenotype is mechanistically distinct from that induced by fasting, denervation, or inactivity, and may instead reflect chronic perturbations in Ca²⁺ homeostasis due to impaired SERCA1 regulation

(Supplementary Fig. 10). Conversely, the soleus muscle, which expresses low levels of MCARE, showed a significant increase in weight in 15-month-old KO mice, likely due to compensatory hypertrophy in response to the atrophy of functionally synergistic muscles such as the gastrocnemius (Fig. 5a). Moreover, loss of *Mcare* led to reduced muscle strength, as KO mice exhibited lower grip strength than WT mice (Fig. 5b). Similar to muscle mass, the decline in grip strength was more pronounced in older age groups. A reduction in muscle mass and strength due to *Mcare* deficiency was observed in both female and male mice (Supplementary Fig. 11a, b). Histological analysis of the quadriceps showed that the distribution of fiber cross-sectional area (CSA) in 6-month-old KO mice appeared to be shifted toward smaller fiber sizes compared with WT mice (Fig. 5c, d). Consistently, the mean CSA was significantly reduced in KO mice. As observed in various muscle diseases, vacuoles develop in some muscle fibers of KO mice (Supplementary Fig. 12a). These were not glycogen granules, as observed in Pompe disease, as they were negative for Periodic Acid-Schiff (PAS) staining (Supplementary Fig. 12b).

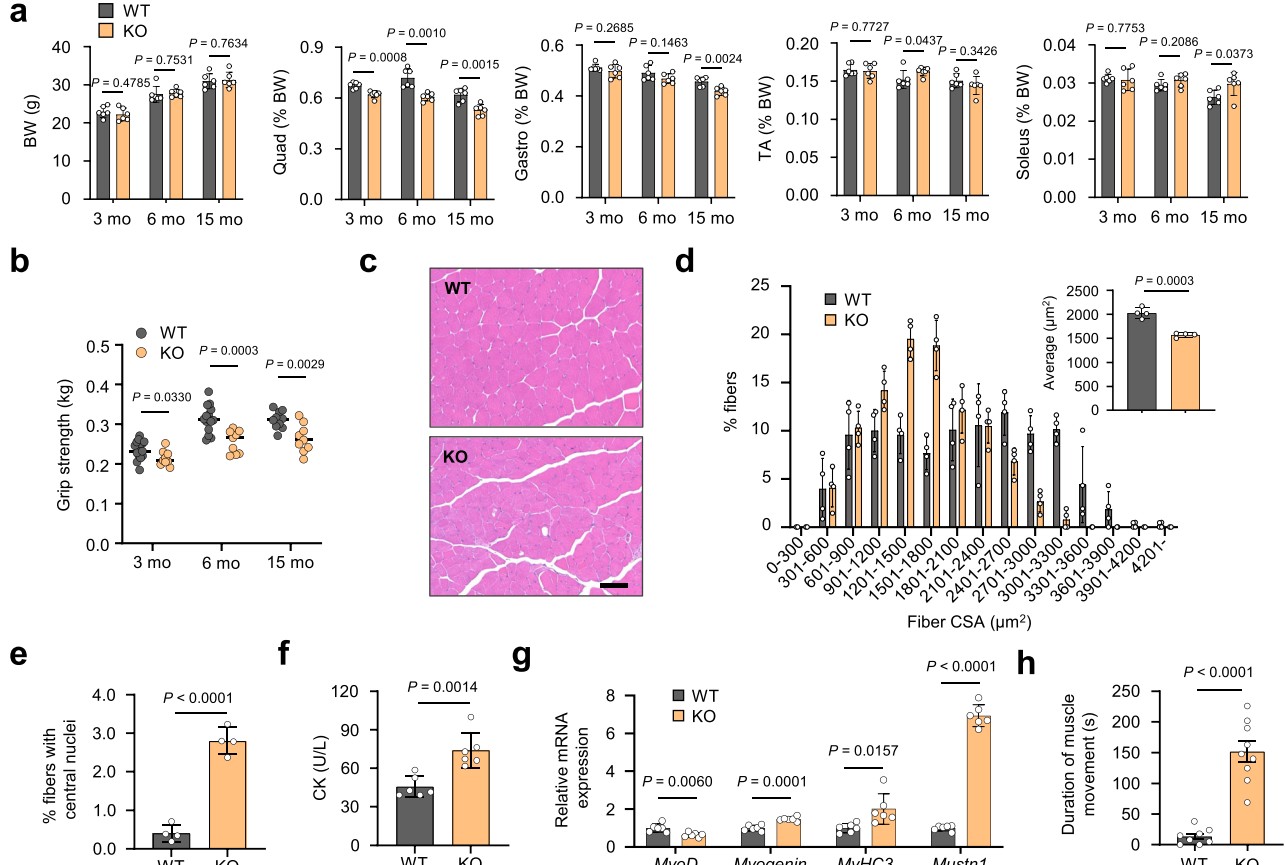

**Fig. 5 | Loss of *Mcare* leads to impaired skeletal muscle function in mice. a** Body weight (BW) and skeletal muscle tissue weight as a percentage of BW in 3-, 6-, and 15-mo mice (mo, months of age). n = 6 mice. **b** Grip strength in 3-, 6-, and 15-mo mice. n = 15 (WT, 3 mo), 9 (KO, 3 mo); 15 (WT, 6 mo), 9 (KO, 6 mo); 10 (WT, 15 mo), 9 (KO, 15 mo) mice. **c** HE staining of the quadriceps from 6 mo WT and KO mice. Scale bar, 100 μm. **d** Frequency distribution and average of cross-sectional area of quadriceps muscle fibers. Each data point represents one mouse. n = 4 mice. **e** Percentage of muscle fibers with central nuclei. n = 4 mice. **f** Serum CK activity in 6 mo WT and KO mice. n = 6 mice. **g** qRT-PCR analysis of relative mRNA expression in quadriceps of WT and KO mice. n = 6 mice. **h** Duration of spontaneous muscle contraction following skin removal. n = 8 (WT) and 9 (KO) mice. Error bars represent mean ± SD. Statistical significance was assessed using a two-tailed unpaired Student's *t*-test.

The appearance of centrally nucleated fibers is a hallmark of muscle regeneration in many degenerative myopathies. In KO mice, we observed a marked increase in such fibers within the quadriceps (Fig. 5e), along with elevated serum CK activity, a widely used marker of muscle damage (Fig. 5f). Gene expression analysis revealed decreased *MyoD* expression and increased expression of *Myogenin*, *MyHC3*, and *Mustn1*−genes typically upregulated during muscle regeneration (Fig. 5g). Together, these histological, biochemical and transcriptional changes indicate ongoing muscle regeneration following damage in KO mice.

Notably, when the skin was detached from the muscles of KO mice, the mechanical stimulus triggered rippling contraction of the muscle (Supplementary Movie 3). This spontaneous muscle movement persisted for an average of approximately 150 s (Fig. 5h). In contrast, WT mice exhibited spontaneous muscle activity lasting only about 14 s on average, which consisted of sporadic, minor spasms rather than the sustained rippling contractions observed in KO mice. Remarkably, the wave-like contractions remained evident even after the hindlimb muscles were completely isolated from the leg (Supplementary Movie 4).

### *Mcare* deficiency increases the vulnerability of skeletal muscle to exercise-induced damage

To further investigate the role of MCARE in skeletal muscle function, we assessed the endurance capacity and exercise-induced molecular and physiological responses of WT and KO mice using two distinct treadmill running protocols. First, an exhaustive treadmill running test

was performed to evaluate endurance capacity. This revealed no significant difference in running distance between the two groups, indicating comparable endurance (Fig. 6a). In a separate experiment, a 30-min submaximal treadmill running protocol was used to assess post-exercise responses. Blood and skeletal muscle samples were collected 60 min after the completion of exercise. Under this condition, serum CK activity was significantly elevated in KO mice, indicating increased muscle damage (Fig. 6b). We next analyzed gene expression in the quadriceps, TA, and soleus muscles. The mRNA levels of *peroxisome proliferator-activated receptor γ coactivator-1a* (*PGC-1a*), a gene typically responsive to exercise, increased similarly in all three muscles in both WT and KO mice, confirming that the exercise load was equivalent across groups (Fig. 6c). In contrast, the stress-inducible *activating transcription factor 3* (*Atf3*) showed a differential pattern: while *Atf3* expression increased in all muscles following exercise, post-exercise *Atf3* levels were significantly higher in KO mice than in WT mice in the quadriceps and TA−fast-twitch muscles where MCARE is abundantly expressed. In the soleus, a predominantly slow-twitch muscle with low MCARE expression, no such difference was observed (Fig. 6c). To determine whether classical stress pathways contribute to the MCARE-dependent increase in *Atf3* expression, we measured the mRNA levels of *Heme oxygenase-1* (*HO-1*), *binding immunoglobulin protein* (*BiP*), and *C/EBP homologous protein* (*Chop*), which are canonical markers of oxidative and ER stress. None of these genes exhibited expression changes consistent with the *Atf3* induction, indicating that these classical stress pathways are unlikely to explain the differential *Atf3* response observed in KO mice (Supplementary Fig. 13). This exclusion

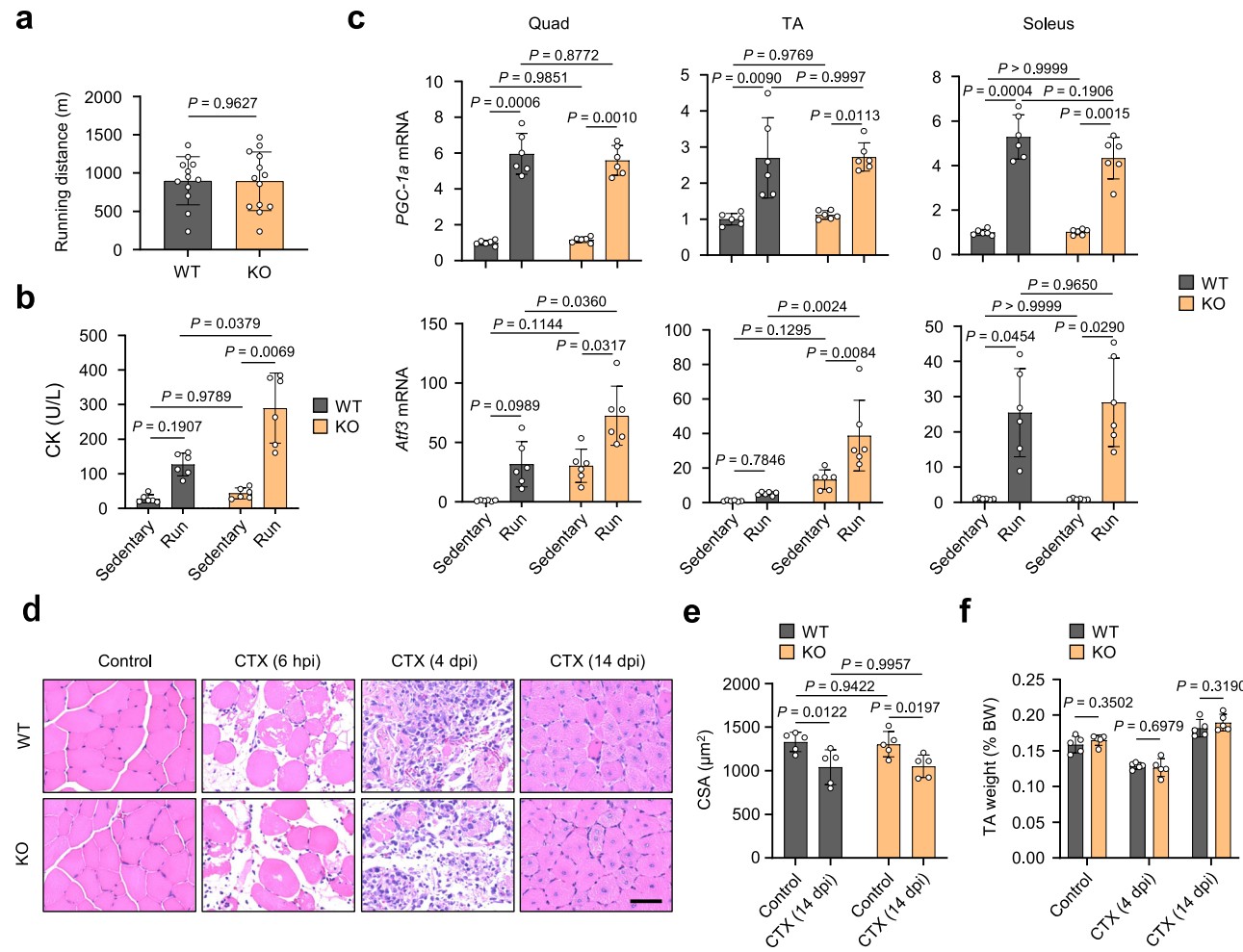

**Fig. 6 | *Mcare* deficiency exacerbates exercise-induced muscle damage but does not impair regeneration. a** Running endurance of 3-month-old WT and KO mice assessed using an exhaustive treadmill running protocol. n = 12 (WT) and 13 (KO) mice. **b** Serum CK activity in WT and KO mice under sedentary conditions and 60 min after a 30-min submaximal treadmill running protocol. n = 6 mice. **c** qRT-PCR analysis of *PGC-1a* and *Atf3* mRNA levels in Quad, TA, and Soleus muscles of WT and KO mice at baseline and 60 min post-exercise. n = 6 mice. **d** HE staining of TA muscle from WT and KO mice injected with CTX. Muscles were collected before CTX injection and at 6 h post-injury (hpi), 4 days post-injury (dpi), and 14 dpi. Scale bar, 50 μm. **e** Mean cross-sectional area of TA muscle fibers. Each data point represents one mouse. n = 5 mice. **f** TA muscle weight (%BW) in uninjured mice and at 4 and 14 dpi in WT and KO mice. n = 5 mice. Error bars represent mean ± SD. Statistical significance was assessed using two-way ANOVA with Tukey's multiple comparisons test.

narrows the possible mechanisms and suggests that the greater *Atf3* induction in KO mice more likely reflects an increased physiological burden on fast-twitch skeletal muscle fibers lacking MCARE, rather than activation of classical oxidative or ER stress responses.

To evaluate the effect of *Mcare* deficiency on muscle regeneration, cardiotoxin (CTX) was injected into the TA muscle, and the progression of muscle regeneration was observed. CTX-injected TA muscle was collected at 6 h post-injury (hpi), 4 days post-injury (dpi), and 14 dpi. A comparison of HE-stained skeletal muscle sections showed no histological differences between WT and KO mice (Fig. 6d), and the reduction in CSA associated with muscle damage was comparable (Fig. 6e). Similarly, the two groups showed no significant difference in TA muscle weight change after CTX injection (Fig. 6f). These data indicate that MCARE had little effect on muscle regeneration after injury, highlighting that the muscle pathology observed in KO mice is likely caused by Ca²⁺ handling abnormalities through mechanisms independent of, rather than due to, defective regeneration.

## Discussion

The SERCA pump is a key regulator of cellular Ca²⁺ homeostasis, facilitating the transport of Ca²⁺ from the cytosol back into the SR after

muscle contraction. SERCAs are the most abundant components of SR membrane proteins, highlighting their essential role in SR function[16].

The present study identified a transmembrane protein, MCARE, that enhances SERCA1 activity specifically in fast-twitch skeletal muscles (Fig. 7). MCARE is predominantly expressed in these muscles, where its distribution aligns with SERCA1 and its inhibitory micropeptide MRLN. MCARE is composed of more than 100 amino acids, excluding it from the micropeptide category, and it features two transmembrane domains, setting it apart from previously reported SERCA regulatory micropeptides. Our results show that MCARE localizes to the SR membrane, enhancing SERCA activity by displacing the MRLN. In line with the established fact that Ca²⁺ reuptake by SERCA is the rate-limiting step in muscle relaxation, MCARE accelerated the relaxation velocity in C2C12 myotubes[17,18]. Complementarily, delayed relaxation in primary myotubes lacking MCARE further supports the role of endogenous MCARE in promoting muscle relaxation. Furthermore, *Mcare* deficiency in vivo induces a dystrophy-like phenotype, including progressive muscle atrophy primarily in the proximal muscles, reduced muscle strength, elevated serum CK levels, and increased skeletal muscle vulnerability. Such pathological features are frequently associated with elevated intracellular Ca²⁺ levels, which are known to

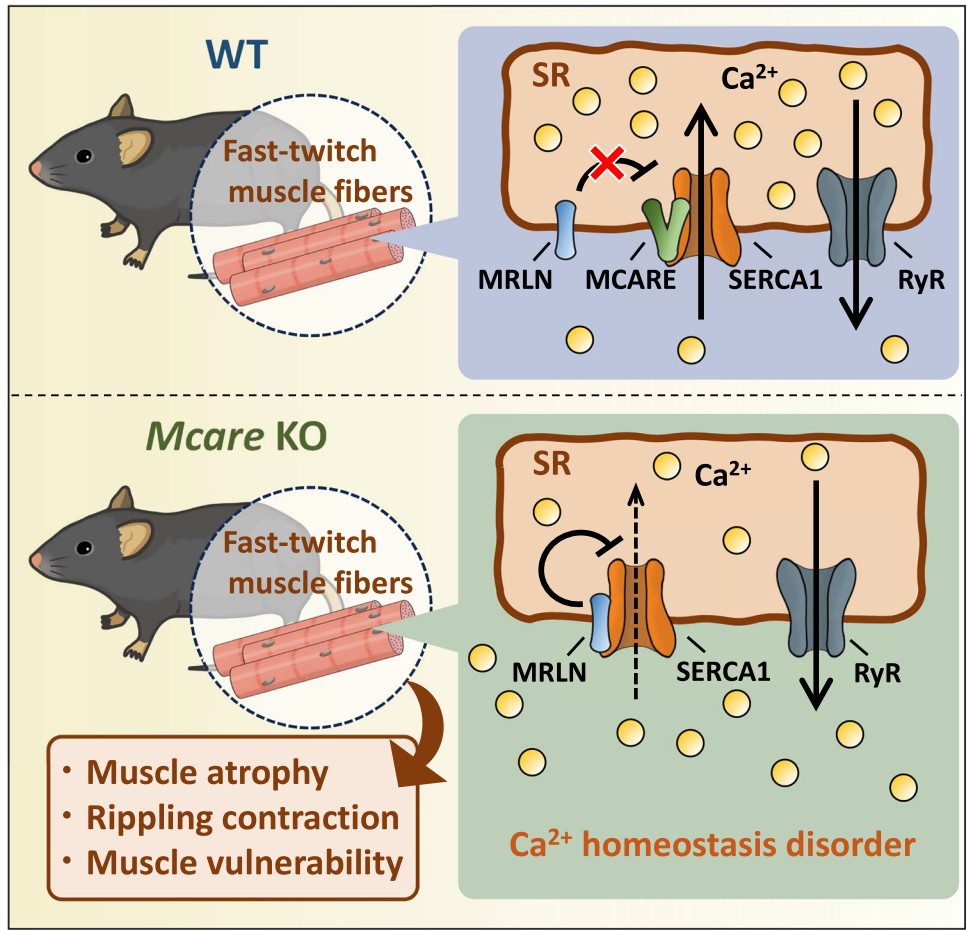

**Fig. 7 | Schematic of MCARE function in fast-twitch skeletal muscles.** In WT muscle fibers, MCARE enhances SERCA1 activity by displacing its inhibitory micropeptide MRLN, thereby maintaining $Ca^{2+}$ homeostasis. In contrast, *Mcare*-deficient fast-twitch fibers exhibit impaired $Ca^{2+}$ reuptake, leading to $Ca^{2+}$ homeostasis disruption. This dysfunction results in muscle atrophy, rippling contractions, and increased vulnerability to physical stress.

contribute to muscle damage and degeneration. For example, intense exercise induces substantial intracellular $Ca^{2+}$ accumulation, leading to exercise-induced muscle damage[19–22]. Moreover, impaired SERCA function due to aging, neurodegenerative diseases, and muscular dystrophies disrupts intracellular $Ca^{2+}$ homeostasis and further contributes to muscle atrophy and weakness[23,24]. Together with these reports, our data indicate that *Mcare* KO mice exhibit disturbances in skeletal muscle $Ca^{2+}$ handling due to impaired SERCA activity, which results in aggravated $Ca^{2+}$-related muscle damage from daily physical activity and running exercise. Significant atrophy in the proximal muscles of KO mice, which are subject to greater mechanical stress during locomotor activity, supports this conclusion. Given its role in exercise-induced muscle damage, calpain—a $Ca^{2+}$-activated protease—may also contribute to the pathological features observed in *Mcare*-deficient muscle[25]. Future studies should explore whether impaired $Ca^{2+}$ homeostasis leads to aberrant calpain activation in this context. In our observation of skeletal muscle cross-sections from KO mice, numerous vacuoles were identified within certain muscle fibers. Morphologically, these vacuoles closely resemble those derived from the T-tubular system owing to eccentric workload or tenotomy-induced muscle damage[26,27]. A previous report suggested that these vesicles can retain $Ca^{2+}$, thereby preventing a rise in cytosolic $Ca^{2+}$ levels and mitigating muscle damage[28]. The vacuoles observed in *Mcare* KO mice may also have formed from the T-tubular system due to muscle injury, potentially attenuating the increase in cytosolic $Ca^{2+}$ concentration. Considering the results of our in vitro and in vivo studies on MCARE, along with its conserved amino acid sequence and muscle-specific

expression in vertebrates, we propose that modulation of SERCA1 activity by MCARE is a fundamental mechanism underlying the contraction-relaxation process and the protection of fast-twitch muscles from $Ca^{2+}$-induced damage.

Although our findings provide important insights into the physiological role of MCARE in skeletal muscle, we acknowledge that the precise molecular mechanisms linking reduced SERCA1 activity to the diverse phenotypes observed in *Mcare* KO mice remain to be fully elucidated. In particular, the connections between impaired $Ca^{2+}$ homeostasis and downstream events such as muscle atrophy, increased susceptibility to damage, and rippling contractions are currently speculative. These observations raise several hypotheses regarding the involvement of calcium-sensitive proteases (e.g., calpains), altered excitation-contraction coupling, or stress-responsive signaling pathways, all of which warrant future investigation using temporally and spatially controlled genetic or pharmacological approaches. Further studies will be necessary to dissect these downstream pathways and determine whether they causally link MCARE deficiency to the observed pathological features. In addition, we recognize that our calcium imaging data, which were primarily obtained using C2C12 myotubes, are subject to limitations due to the absence of complex SR and neuromuscular junction architecture in these cells.

Muscle fibers are generally classified into two primary types: slow-twitch (type I) and fast-twitch (type IIa/IIx/IIb). Fast-twitch fibers have a more developed SR, resulting in enhanced $Ca^{2+}$ handling and faster contraction-relaxation velocities[29]. Among these, type IIb fibers are

highly glycolytic, containing minimal mitochondria and specializing in brief, explosive power and maximum force production. A previous report using single-nucleus RNA-seq of myonuclei isolated from mouse skeletal muscle showed that *Tmem233/Mcare* is enriched in *MyHC4*-positive myonuclei, which originate from type IIb muscle fibers[30]. Additionally, a decrease in *Tmem233/Mcare* gene expression has been observed in the skeletal muscle of large Mafs triple KO mice, which display a significant reduction in type IIb muscle fibers[31]. These reports suggest that MCARE is specifically expressed in type IIb muscle fibers. Given that SERCA1 is expressed in all fast-twitch muscle fibers, MCARE is not essential for SERCA1 function but may be an important factor in characterizing type IIb fibers.

A distinctive phenomenon observed in *Mcare* KO mice is the rippling contraction of skeletal muscle, induced by stretching stimuli during skin removal. This rippling occurred after cervical dislocation and persisted in completely detached muscle tissue, indicating its independence from motor neuron action potentials. Interestingly, these features closely resembled the symptoms of rippling muscle disease (RMD) patients. RMD is a benign myopathy with wave-like muscle contractions and mounds induced by percussion or stretching. Electromyography demonstrates that abnormal muscle contractions caused by RMD are electrically silent, indicating that action potentials are not involved[32]. RMD is often associated with limb-girdle muscular dystrophy type-1C, which is caused by mutations in the caveolin-3 (Cav3) gene[33,34]. While the precise mechanism remains unclear, several reports have indicated that pathogenic mutations or defects in Cav3 lead to T-tubule disorganization and disrupted $Ca^{2+}$ homeostasis. This suggests that the distinctive contractions observed in RMD may result from $Ca^{2+}$ dysregulation within the muscle fibers[35–37]. To date, no animal model that reproduces the rippling muscle contractions characteristic of RMD has been reported, which has significantly impeded research on the disease. For instance, although *Cav3* KO mice exhibit mild myopathic changes and abnormalities in the T-tubule system, rippling muscle contractions have not been observed in these models[38]. In contrast, *Mcare* KO mice display clear rippling muscle contractions in response to mechanical stimulation, even in the absence of motor neuron input. This observation supports the hypothesis that disturbances in intracellular $Ca^{2+}$ homeostasis are sufficient to cause this phenomenon. While further research is needed to determine the extent to which the rippling in *Mcare* KO mice mimics RMD, these mice may represent a useful animal model that enables investigation into the pathomechanisms of this disease.

Recently, it has been reported that *Tmem233/Mcare*, expressed in dissociated dorsal root ganglion (DRG) neurons, interacts with voltage-gated sodium ($Na_V$)1.7 channels on the cell membrane, which play a central role in pain transmission[39]. This interaction is required for Excelsatoxin A (ExTxA), a pain-inducing knottin peptide from the Australian stinging tree (*Dendrocnide excelsa*), to block $Na_V$1.7 inactivation. Notably, TMEM233/MCARE exhibits differential intracellular localization and protein interactions in skeletal muscle cells compared with DRG neurons. This suggests that the function of MCARE in skeletal muscle is likely independent of its interaction with $Na_V$ channels. While $Na_V$1.7 binding with TMEM233/MCARE is crucial for mediating the effects of ExTxA, TMEM233/MCARE has minimal influence on $Na_V$1.7 function in the absence of ExTxA. *Tmem233*cre/cre KO mice exhibited no alterations in nociceptive responses to mechanical, heat, or cold stimuli, suggesting an unclear role for TMEM233/MCARE in DRG neurons under physiological conditions[40]. Given these reports and the significant changes we observed in the skeletal muscle of *Mcare* KO mice, it is evident that, while MCARE may be expressed in non-skeletal muscle cells, its function is particularly critical in skeletal muscle physiology. Nonetheless, it remains uncertain whether the phenotypes observed in *Mcare* KO mice are entirely attributable to MCARE in skeletal muscle cells, necessitating further investigation.

In summary, our results demonstrate that MCARE is a previously uncharacterized SERCA1 regulatory protein that supports $Ca^{2+}$ handling and contributes to the resilience of fast-twitch skeletal muscles to exercise-induced damage. Deletion of *Mcare* results in muscular dystrophy-like features as well as rippling contractions that resemble those observed in RMD. These RMD-like contractions raise the possibility that *Mcare*-deficient mice may serve as a new animal model for studying this rare disorder. These findings underscore the critical role of fiber-type-specific regulators of SERCA activity in maintaining muscle integrity and provide a foundation for future investigation into the contribution of MCARE to skeletal muscle disease pathogenesis.

## Methods

### Plasmid construction

Mouse *Mcare*, *SERCA1a*, *SERCA2*, and *MRLN* cDNAs were amplified from skeletal muscle cDNA and subcloned into pcDNA3.1(+), p3×FLAG-CMV7.1, p3×FLAG-CMV-14, pEGFP-C1, or pmCherry-C1 vectors depending on the desired tag. For constructs carrying an N-terminal or C-terminal Myc-tag, the sequence GAGCAGAAGCTGATCTCAG AGGAGGACCTG was incorporated into the forward or reverse primer, respectively. All plasmid sequences were verified by Sanger sequencing.

### Cell culture

C2C12 (Cat. No. CRL-1772) and HEK293 (Cat. No. CRL-1573) cells were obtained from the ATCC. The cells were cultured in Dulbecco's modified Eagle's medium (DMEM) supplemented with 10% fetal bovine serum (FBS), 100 U/mL penicillin, and 100 µg/mL streptomycin. To induce myogenic differentiation, C2C12 myoblasts were cultured in differentiation medium (DMEM supplemented with 2% horse serum, 100 U/mL penicillin, and 100 µg/mL streptomycin) for at least 4 days.

Satellite cells were isolated from the skeletal muscles (quadriceps and gastrocnemius) of 3-week-old male mice using modifications of previously described methods[41]. In brief, the muscle tissue was stripped of connective tissue and fat and then further minced using a razor blade. The tissue was enzymatically digested with 2.0 mg/mL collagenase type II (Worthington) and 500 PU/mL dispase II (Fujifilm Wako Chemicals) in DMEM for 60 min in a 37 °C water bath with shaking to promote satellite cell release. The tissue was then passed through an 18G needle 2–3 times and incubated for another 30 min. Following digestion, the tissue slurry was centrifuged at $1000 \times g$ for 5 min, and the digestive enzymes were discarded as the supernatant. 10 mL of growth medium (DMEM supplemented with 20% FBS, 100 U/mL penicillin, 100 µg/mL streptomycin, 0.5% chick embryo extract (Usbio), and 100 ng/mL bFGF (R&D Systems)) was added to the cell pellet. After suspending the cells, they were sequentially passed through 100-µm and 40-µm cell strainers. The filtered cells were plated in collagen-coated 6-well plates and incubated for 2 h. The supernatant containing non-adherent cells was transferred into a new collagen-coated 6-well plate. After 24 h of incubation, the supernatant was transferred into a 15-mL tube and centrifuged at $900 \times g$ at 4 °C for 5 min. The supernatant was discarded, and the resulting cell pellet was resuspended in 5 mL growth medium. The cells were suspended in a collagen-coated 6-well plate and incubated. The pre-plate technique described above was repeated twice. The satellite cells obtained were maintained in growth medium. To induce myogenic differentiation, satellite cells at 80% confluence were switched to differentiation medium (DMEM supplemented with 5% horse serum, 100 U/mL penicillin, and 100 µg/mL streptomycin) for 5 days. Cells were maintained in a 5% $CO_2$ incubator at 37 °C.

### Magnetic-activated cell sorting (MACS)

Total mononuclear cells (MACS total), muscle satellite cells (MSCs), and macrophages from WT mice were isolated through MACS (Miltenyi Biotec). To obtain sufficient quantities of MSCs and macrophages

for analysis, muscle tissue was injured by CTX 3 days before cell isolation. Muscle tissues were washed three times with ice-cold PBS and minced with a razor blade. The minced tissue was weighed and resuspended in an enzymatic solution (PBS containing 2 mg/mL collagenase (Worthington Biochemical Corporation), 400 PU/mL dispase II (Fujifilm Wako Chemicals), and 137 mM NaCl). Tissue digestion was performed in a 37 °C water bath with gentle shaking for 1 h, followed by three passes through an 18G needle and an additional 30-min incubation. The suspension was then filtered through 100 μm and 40 μm cell strainers (Corning), and centrifuged at $300 \times g$ for 5 min at 4 °C. The resulting pellets were resuspended in 100 μL of MACS buffer (PBS supplemented with 1 mM EDTA and 0.5% bovine serum albumin (Fujifilm Wako Chemicals, BSA)). Red blood cells were lysed using an RBC lysis buffer (Miltenyi Biotec) for 10 min at room temperature (RT). After centrifugation at $300 \times g$ for 5 min at 4 °C, pellets were diluted in MACS buffer (MACS total). Fc receptor (FcR) blocking was performed by adding FcR blocking reagent (Miltenyi Biotec) to the cell suspension according to the manufacturer's instructions, and incubating for 10 min at 4 °C. For MACS, a 20-fold MACS buffer was added to the cell suspension and centrifuged at $800 \times g$ for 5 min at 4 °C. Magnetic bead-conjugated antibodies were used to select cells according to the manufacturer's protocol. A hematopoietic lineage cell detection cocktail (Miltenyi Biotec, Cat. No. 130-092-613), anti-CD31 antibody (Miltenyi Biotec, Cat. No. 130-119-662), and anti-CD140a antibody (Miltenyi Biotec, Cat. No. 130-101-905) were used for the negative selection. Macrophages were isolated using anti-CD11b MicroBeads (Miltenyi Biotec, Cat. No. 130-049-601), and MSCs were isolated using a biotin anti-integrin-α7 antibody (Miltenyi Biotec, Cat. No. 130-101-979).

### Differential permeabilization method for topology determination

C2C12 myoblasts transfected with Myc-SERCA1a and Flag-MCARE were fixed with 4% paraformaldehyde (Fujifilm Wako Chemicals) for 15 min. The cells were selectively permeabilized with 0.005% digitonin (Nacalai Tesque) for 10 min or completely permeabilized with 0.5% Triton X-100 (Fujifilm Wako Chemicals) for 5 min. The cells were subsequently blocked with blocking buffer (PBS containing 3% BSA) for 60 min at RT, and incubated with the primary antibodies diluted in blocking buffer for 60 min at RT. The primary antibodies used in this study were anti-FLAG M2 antibody (1:500, Sigma-Aldrich, Cat. No. F3165) and anti-c-Myc antibody (1:500, Sigma-Aldrich, Cat. No. C3956). The cells were then incubated with secondary antibodies, a combination of Alexa Fluor 568 conjugated anti-rabbit IgG (1:500, Thermo Fisher Scientific) and Alexa Fluor 488 conjugated anti-mouse IgG (1:500, Thermo Fisher Scientific) for 30 min at RT in the dark. The glass slides were mounted using VECTASHIELD mounting medium with DAPI (Vector Laboratories) and imaged under LSM800 with Airyscan (Zeiss).

### Quantitative real-time PCR (qRT-PCR)

Total RNA was extracted from C2C12 myotubes and mouse tissues using ISOGEN (NIPPON GENE) according to the manufacturer's instructions. A high-capacity cDNA reverse transcription kit (Applied Biosystems) was used to synthesize and amplify cDNA from the total RNA. Quantitative real-time PCR was performed using an Applied Biosystems StepOnePlus instrument. For tissue samples, each mouse was considered one biological replicate, and for cell culture experiments, each well containing independently cultured myotubes was treated as one biological replicate. The 18S rRNA levels were used as an internal control to normalize the mRNA levels of each gene. The primers used for PCR analysis are listed in the Supplementary information (Supplementary Table 1).

### Western blot

Cells and mouse skeletal muscle were lysed in RIPA buffer (50 mM Tris-HCl (pH 8.0) containing 150 mM NaCl, 1% Triton X-100, 0.5% deoxycholate, and 0.1% SDS) supplemented with a protease inhibitor cocktail (PIC, Nacalai Tesque) and a phosphatase inhibitor cocktail (Sigma-Aldrich). The lysates were subjected to SDS-PAGE and transferred to polyvinylidene difluoride membranes (Millipore, Billerica, MA). The membrane was blocked with 5% BSA (Fujifilm Wako Chemicals) for 60 min at RT and incubated overnight at 4 °C with primary antibody, anti-MCARE antibody (1:1,000, generated by immunizing rabbits with the peptide CPEAYTEDDKTEED), anti-Caveolin-3 antibody (1:200, Santa Cruz Biotechnology, Cat. No. sc-5310), anti-N-Cadherin antibody (1:1000, Cell Signaling Technology, Cat. No. 4061), anti-Lamin A/C antibody (1:1000, Cell Signaling Technology, Cat. No. 2032), anti-Tom20 antibody (1:1000, Cell Signaling Technology, Cat. No. 42406), anti-GM130 antibody (1:1000, BD Bioscience, Cat. No. 610823) and anti-HSP90 antibody (1:200, Santa Cruz Biotechnology, Cat. No. sc-13119), or incubated for 60 min at RT with other primary antibodies, anti-RyR antibody (1:300, Santa Cruz Biotechnology, Cat. No. sc-376507), anti-SERCA1 antibody (1:1000, Cell Signaling Technology, Cat. No. 4219), anti-SERCA2 antibody (1:200, Santa Cruz Biotechnology, Cat. No. sc-376235), anti-FLAG M2 antibody (1:2000, Sigma-Aldrich, Cat. No. F3165), anti-Myc antibody (1:1000, Sigma-Aldrich, Cat. No. C3956), anti-GFP antibody (1:1000, Cell Signaling Technology, Cat. No. 2956) and anti-GAPDH antibody (1:5000, Proteintech. Cat. No. 10494-1-AP). The membranes were incubated with secondary antibodies for 60 min at RT. Chemiluminescent signals were determined using a Fusion Solo S (Vilber).

### Microsome preparation from skeletal muscle

Quadriceps muscles were collected from adult mice and homogenized in homogenization buffer (250 mM sucrose, 10 mM Tris-HCl (pH 7.4), 1 mM EDTA, and PIC (Nacalai Tesque)) using a Dounce homogenizer. The homogenate was centrifuged at $1000 \times g$ for 10 min at 4 °C to remove nuclei and cell debris. The resulting supernatant was centrifuged at $10,000 \times g$ for 15 min at 4 °C to remove the mitochondrial fraction. The supernatant was then subjected to a second $10,000 \times g$ spin for 10 min and collected. To isolate the microsomal fraction, the supernatant was ultracentrifuged at $100,000 \times g$ for 60 min at 4 °C. The pellet was washed with homogenization buffer and resuspended in RIPA buffer supplemented with PIC.

### Co-immunoprecipitation (Co-IP) assay

Cells were lysed in Co-IP buffer (20 mM $NaPO_4$, 150 mM NaCl, 2 mM $MgCl_2$, 0.1% NP-40, and 10% glycerol; pH 7.4) supplemented with protease inhibitor cocktail (Nacalai Tesque), phosphatase inhibitor cocktail (Sigma-Aldrich), and 1 mM dithiothreitol. Total cell lysates containing 500 μg of protein were incubated with anti-DDDDK-tag mAb-magnetic agarose (MBL) for 60 min at 4 °C with gentle rotation. The beads were washed four times with Co-IP buffer and eluted with TBS buffer containing 150 μg/mL 3× FLAG peptide (Protein Ark). The eluted samples were analyzed by Western blot.

### Adenovirus generation and transduction

*MCARE* was amplified by PCR from human skeletal muscle cDNA and subcloned into a pENTR1A-Dual selection vector (Thermo Fisher Scientific). It was then cloned into the pAd/CMV/V5/DEST vector using LR Clonase II from Gateway technology systems (Thermo Fisher Scientific). The plasmid was digested with Pac I (New England Biolabs) and transfected into HEK293A cells to obtain the MCARE expression adenovirus solution. The adenovirus was amplified by infecting HEK293A cells, and the titer was determined.

For transduction, C2C12 myotubes, differentiated for 3 days, were incubated with $3.0 \times 10^6$ plaque-forming units/mL adenovirus medium for 15–18 h. Following incubation, the cells were washed three times with PBS and further cultured in differentiation medium. Infected C2C12 myotubes were used for various experiments 36–48 h post-infection.

## Measurement of SERCA activity

HEK293 cells were transfected with a SERCA1a expression vector along with MRLN, MCARE expression vectors, or both. For MCARE, a three-fold excess of the expression vector was used compared to MRLN. After 36 h, the cells were harvested in calcium-free PBS and washed with PBS containing 5 mM EDTA, followed by two additional washes with calcium-free PBS. The cells were then centrifuged at $900 \times g$ for 6 min, and the resulting pellets were incubated on ice for 10 min in a hypotonic buffer (10 mM Tris-HCl (pH 7.5) supplemented with 0.5 mM $MgCl_2$) containing a protease inhibitor cocktail. The cells were homogenized using a Dounce homogenizer and diluted with an equal volume of homogenate dilution buffer (10 mM Tris-HCl (pH 7.5) containing 0.5 M sucrose, 6 mM 2-mercaptoethanol, 40 μM $CaCl_2$, and 300 mM KCl). The suspension was centrifuged at $10,000 \times g$ for 20 min, and the supernatant was collected. KCl was added to a final concentration of 0.6 M, and the mixture was centrifuged at $100,000 \times g$ for 60 min. The resulting pellets were resuspended in microsome suspension buffer (20 mM MOPS-KOH (pH 7.0) containing 250 mM sucrose and 1 mM $MgCl_2$) and centrifuged again at $100,000 \times g$ for 60 min. The final pellets were resuspended, and protein concentrations were equalized across samples. Western blot confirmed equivalent SERCA1 expression levels in the samples.

ATPase activity was assessed in a 39.5 μL reaction mixture (40 mM MOPS-KOH (pH 7.0) containing 100 mM KCl, 5 mM $NaN_3$, 5 mM $MgCl_2$, 0.1 mM ouabain, and 1 mM EGTA) containing 3 μL of microsome solution, 2.5 μL of various concentrations of $CaCl_2$, and 5 μL of reaction starter solution (100 μM A23187, 40 mM ATP) at 37 °C. The reactions were terminated at 5 and 25 min by adding 25 volumes of 0.06% TCA. The amount of phosphate released was measured using the malachite green phosphate assay kit (BioAssay Systems). ATPase activity was determined as the difference between phosphate levels at 25 and 5 min. Each experiment was independently repeated four times using separately transfected HEK293 cell cultures, and the data represent mean values from four biological replicates.

## Muscle contraction motion analysis

C2C12 myotubes differentiated for 5 days were subjected to 23 V, 2 ms, 0.5 Hz EPS for 18 h using a cell culture stimulator (C-PASE EP, Ion Optix) to acquire contractile capacity. EPS was continued under the same conditions, and movies were recorded under a microscope (BZ-X810 system, Keyence). Motion analysis was performed using a motion analyzer (BZ-X800 analyzer, version 1.0.0.2; Keyence) to calculate the distance and velocity of a selected point on a myotube moving along a specified straight line. Movies were captured from nine or ten different fields of view obtained from at least three independent wells per condition, and three myotubes showing the strongest visible contractions in each field were selected for analysis. Myotubes that did not exhibit any observable contraction in response to EPS were considered non-responsive and were excluded from the analysis. Motion analysis was conducted in differentiation medium, which was maintained throughout the stimulation and imaging process.

## Intracellular calcium imaging

C2C12 myotubes infected with adenovirus expressing LacZ or MCARE and primary myotubes from WT and Mcare KO mice were washed twice with PBS and then incubated with Fluo-8 loading medium (DMEM containing 10 μM Fluo-8 AM (Abcam) and 0.1% Cremophor EL (Sigma)) at 37 °C. After 1 h of incubation, the myotubes were washed twice with PBS and then replaced with FluoroBrite DMEM (Thermo Fisher Scientific) supplemented with 10% FBS and 4 mM L-glutamine, in accordance with the manufacturer's recommendation for optimal fluorescence imaging. The medium was pre-equilibrated in a 5% $CO_2$ incubator at 37 °C for at least 30 min before use, and after replacement, the cells were further incubated for 5 min to stabilize pH and temperature prior to imaging. The cells were then stimulated with electric pulses of 23 V for 2 ms or 6 ms. The intracellular calcium responses were monitored using an all-in-one fluorescence microscope (BZ-X810 system, Keyence). All imaging procedures were completed within 5 min after removing the plate from the incubator, which minimizes pH fluctuations under $CO_2$-independent conditions. For each field of view in the captured movie, two or three myotubes that responded to electrical stimulation were randomly selected, and changes in their fluorescence intensities were quantified using Fiji (ImageJ, version 2.14; National Institutes of Health). This analysis was repeated for at least six fields of view obtained from three independent wells per condition.

## Mice and diet

Male wild-type C57BL/6J mice were purchased from CLEA Japan, Inc. for experiments using only wild-type animals, whereas wild-type littermates were used as controls when Mcare KO mice were employed. Mice were housed in an animal care facility under controlled temperature and humidity conditions with a 12-h light/dark cycle. The mice were given free access to water and diet (Labo MR Stock; Nosan Corporation Bio Department). Each mouse was considered one biological replicate, and "n" refers to the number of individual mice used per group.

## Generation of Mcare KO mice

Three single-guide RNAs (sgRNAs) targeting the Mcare gene with no homologous sequences on the other genes within 20 bases upstream of the protospacer adjacent motif (PAM) (5′-GTACAGAAGATGACAA-GACCGAGG-3′, 5′-GTGGGCAGAAACACGAGATAATGG-3′, and 5′-GCGATGTTGACCGGGTACGCTGGG-3′), were selected using CRISPR-direct (http://crispr.dbcls.jp/). Oocytes collected from superovulated C57BL/6J female mice (CLEA Japan) were fertilized with C57BL/6J sperm in vitro. Cas9/gRNA ribonucleoprotein complex was delivered into embryos by electroporation and transferred into the oviducts of 0.5-day-post-coitum ICR recipients (Charles River Laboratories JAPAN) to obtain heterozygous mutant mice.

## Muscle electroporation

Male C57BL/6J mice were anesthetized with isoflurane, and the TA muscle was injected with 50 μL of 40 UI/mL hyaluronidase solution (Irvine Scientific) 1 h before plasmid injection. Then, 60 μL of plasmid solution containing 15 μg each of EGFP-MCARE and mCherry-SERCA1a was injected, and the muscle was immediately stimulated with three consecutive electroporations (three pulses of 70 V for 50 ms with an interval of 50 ms) at 3 s intervals using the electroporator NEPA21 (NepaGene) and a parallel fixed 5 mm gap needle (CUY560-5-0.5, NepaGene). Five days later, the mice were sacrificed, and the TA muscles were harvested.

## Treadmill exercise

Three-month-old male WT and KO mice were subjected to two distinct treadmill exercise protocols to evaluate endurance and post-exercise responses. All mice were first acclimated to treadmill running (Muromachi Kikai) at a 10% incline. For endurance testing, the treadmill speed was initially set at 5 m/min and increased by 5 m/min every 5 min up to a maximum of 20 m/min. Running was continued until mice reached exhaustion, defined as remaining on the electric stimulus grid for more than 7 s without re-engaging with the treadmill belt. For submaximal exercise testing, mice were run for a fixed duration of 30 min using the same incremental protocol as in the endurance test: treadmill speed was initially set at 5 m/min and increased by 5 m/min every 5 min up to a maximum of 20 m/min. Mice that completed the full session without reaching the exhaustion criterion were considered to have successfully completed the protocol. Blood and skeletal muscle tissue samples were collected 60 min after the end of exercise for subsequent molecular and biochemical analyses.

## Grip strength test

The grip strength test was performed using a grip strength meter (MK-380M, Muromachi Kikai). Grip strength was measured five times for each mouse. The same measurements were repeated 4 days later, and the highest recorded values from each session were averaged for analysis. The test was performed on both male and female mice. Male mice were tested at 3, 6, and 15 months of age, and female mice were tested at 6 months of age. Detailed sample sizes for each group are provided in the corresponding figure legend.

## Respiratory gas analysis

$O_2$ consumption and $CO_2$ production in 3-month-old male WT and KO mice were measured using an ARCO-2000 Mass Spectrometer (ARCO System) with one mouse per chamber (WT, n = 5; KO, n = 4). Energy expenditure and RER were calculated based on BW, $VO_2$, and $VCO_2$. The environment was maintained at $21 \pm 3°C$, with $50 \pm 10\%$ relative humidity. The mice were given free access to water and diet (Labo MR Stock; Nosan Corporation Bio Department).

## CTX-induced muscle injury

Male, 3-month-old WT and KO mice were used for CTX injection experiments. CTX (LATOXAN) was diluted to a final concentration of 10 μM in PBS and injected into the TA muscle in a 50-μL volume. TA muscles were then collected at the indicated time points (n = 5 per group).

## Muscle fiber cross-sectional area analysis

Quadriceps or TA muscles from male WT and KO mice were fixed in 10% formalin neutral buffer solution (Fujifilm Wako Chemicals) for more than 24 h, and embedded in paraffin. Section preparation and HE staining were performed by GenoStaff (https://genostaff.com/). Digital images of HE-stained mid-belly sections of the quadriceps and TA muscles were captured and analyzed for muscle fiber cross-sectional area using a BZ-X800 analyzer (Keyence), with one section analyzed per muscle for each mouse. More than 200 fibers in the quadriceps muscle and 120 fibers in the TA muscle were traced per sample. Sample sizes are provided in the legends.

## Statistics and reproducibility

Statistical significance was determined by two-tailed unpaired Student's t-test or two-tailed two-way ANOVA with Tukey's multiple comparisons test performed using GraphPad Prism (version 9.5.1; GraphPad Software, LLC). To determine the effects of MCARE on maximum contraction and relaxation velocities adjusted for contraction displacement, ANCOVA was performed using SAS OnDemand for Academics (SAS Institute Inc.). Even when a significant interaction between contraction or relaxation velocity and contraction displacement was detected, the overall ANCOVA results were presented as reference values for comparison. All key experiments, including microscopy and immunoblot analyses, were independently repeated at least three times with similar results. Quantitative data are reported as the mean ± SD, derived from at least three independent biological replicates. Details regarding the number of experiments, sample size, and statistical tests are provided in the corresponding figure legends. Although the data distribution was assumed to be normal, it was not formally tested. No statistical methods were employed to predetermine the sample size, and no data were excluded from the analyses. Statistical significance was defined as $P < 0.05$.

## Ethical approval

All animal experiments were conducted in accordance with the relevant ethical regulations and were approved by the Animal Usage Committee of the University of Tokyo under protocol numbers P19-016 and P22-125.

## Reporting summary

Further information on research design is available in the Nature Portfolio Reporting Summary linked to this article.

## Data availability

All data supporting the findings of this study are available in the Source Data file. Source data are provided with this paper.

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

## Acknowledgements

This work was supported by the Lotte Shigemitsu Prize from the LOTTE Foundation (to T.S.), the Japan Society for the Promotion of Science (JSPS) KAKENHI Grant JP19K15786 (to T.S.), JP23K05089 (to T.S.), and JP20H00408 (to R.S.).

## Author contributions

T.S. conceived the project, designed and performed experiments, conducted data analysis, interpretation, and statistical analysis, supervised the research, and wrote the manuscript. H.T. performed MACS experiments. T.S., M.K., and A.A. generated the *Mcare* KO mice. Y.T. and Y.Y. conducted data analysis and discussed the results. R.S. designed the overall study and experiments, edited the manuscript, and supervised the project team.

## Competing interests

The authors declare no competing interests.
