## [Transparent Peer Review file · Nature Communications]

MCARE enhances SERCA1 activity in fast-twitch muscle to maintain calcium handling and muscle integrity

Corresponding Author: Dr Takashi Sasaki

Version 0:

Reviewer comments:

Reviewer #1

(Remarks to the Author)

In the manuscript by Sasaki et al. entitled "MCARE is a SERCA activator in fast-twitch skeletal muscle", the authors identified the role of the protein MCARE in skeletal muscle. By using in vitro experiments and generating knockout mice, they showed that MCARE is specifically expressed in fast skeletal muscle and interacts with Serca1 to activate the reuptake of cytoplasmic calcium into the sarcoplasmic reticulum.

Overall, this is a comprehensive study with convincing results, the design of the experiments is good, and the conclusions are solid. The article is well written, and I favor the publication of this article in the Nature Communication journal without significant revision.

However, there are minor suggestions that could improve the article:

-In line 120, the authors could add a simple scheme in supplemental data to show the model of MCARE structure in the SR membrane.

-In line 182, the authors could again add a scheme in supplemental data showing the MCARE gene (with exon and intron) and the localization of the sgRNA used to generate the MCARE knock out mouse.

-In line 191, this phenotype could be better highlighted. Could these data have been presented in the main figure and not just as a supplementary video? For example, pictures of WT and KO muscle over time after dissection could be presented. Or measuring the continued contraction duration.

Reviewer #2

(Remarks to the Author)

Ca²⁺ signaling has long been recognized as a regulator of muscle contraction through the process of excitation contraction coupling (ECC). ECC involves the dynamic release and uptake of Ca²⁺ from the sarcoplasmic reticulum (SR). Contraction results when Ca²⁺ is released through the RYR channels located in the SR membrane and then Ca²⁺ binds to the myofibril apparatus to stimulate action-myosin ATPase activity. Relaxation involves re-sequestration of Ca²⁺ from the myofibril to the SR by the SERCA pump. Details of the SERCA activity is therefore of great interest in understanding muscle physiology and contractility.

In the current work, the authors have identified a novel transmembrane protein TMEM233 as a regulator of SERCA function that has been renamed MCARE. MCARE therefore joins a growing number of micropeptides that regulate SERCA function. Myoregulin, sarcolipin and phospholamban act as endogenous inhibitors of SERCAs. Their different patterns of tissue expression for these micropeptides confers differential regulation of the specific SERCA. In contrast DWORF, Neuronatin and STIM1 have been shown to restrict access of these micropeptides to SERCA and thereby modulates SERCA activity. In the current work the authors propose that MCARE acts in a similar manner to prevent MLN access to the SERCA1 in fast

twitch muscles. The work is significant, if validated, and offers new and important information of Ca²⁺ signaling in muscle.

1) Prior studies of TMEM233 identified it as an interacting partner for Nav1.7 where it can modulate gating properties of sodium channels. It has also been shown to be the target of the toxin ExTxA. Given the key role of Na channels in muscle excitability (particularly for EPS) how do the authors reconcile their data with the recent publication by Jami et al (PMID: 37117223)? This seems like a key issue. The authors stimulate cells with electrical pulses (EPS). It is possible that MCARE and Nav channels interaction may explain results of altered Ca²⁺ transients. Can the authors utilize the toxin to test their hypothesis. Does blocking Nav channels in muscle influence MCARE actions?

2) The main Ca²⁺ measurements made in this manuscript involve gain and loss of MCARE in C2C12 cells. These cells are immature muscle, lack motor nerve innervation, and often don't form fully functional SR and T-tubules which may influence the interpretation of the present data. Ideally these experiments are done with isolated fast twitch fibers from mice. Second, more extensive studies would be required to investigate the effects on SERCA1 especially since C2C12 cells express similar levels of both SERCA1 and SERCA2a.

3) The authors show that overexpressing MCARE in C2C12 cells result in dramatic reduction in the Ca²⁺ signal (Fig 2 b-C) whereas myotubes from MCARE KO mice have gain in the Ca²⁺ released (Fig 4 b-C). Ca²⁺ transients represent the summation of events that include Ca²⁺ flux across the plasma membrane and Ca²⁺ flux into and out of the SR. Many factors including SERCA1 are involved. The present studies focus on changes to Ca²⁺ release of SR stores, but do not consider the other issues like Ca²⁺ exchange/transport across the PM and Ca²⁺ leak from the SR. It would be better to design protocols that focus on Ca²⁺ uptake. More detailed studies would be needed to demonstrate the specific Ca²⁺ affect. SERCA1 function is mainly influenced by the terminal portion of the Ca²⁺ transient and is usually assessed as the tau. Authors should consider direct SR Ca measurements with GECI like D1ER or Flou5N. Permeabilized membranes of fibers enable control of Ca²⁺ concentration, ATP availability and SERCA1 activity.

4) If the MCARE activates SERCA1 function, through binding MRLN then overexpressing MCARE would increase the SR Ca²⁺ stores resulting in greater release following EPS. In contrast, if deletion of MCARE results in constitutive interaction of MRLN and reduce the SR Ca²⁺ stores. The studies in Figure 2 and 4 show the opposite raising concerns about the authors interpretation that MCARE activates SERCA1. It is important to show that in Figures 2 and 4 that MRLN interaction with SERCA1 is affected by the gain and loss of function of MCARE similar to studies in figure 3 (all from 293 cells).

5) The studies in figure 5 address the response to injury that is associated with exercise. PGC1 and ATF3 levels are increased and exaggerated in the KO mice. How does this relate to the changes in Ca²⁺ flux? Or SERCA1 ATP consumption? Seems like Ca²⁺ levels at the contractile apparatus would not clear efficiently if MCARE activates SERCA1. What happens to calpain activity?

Reviewer #3

(Remarks to the Author)

This manuscript identified transmembrane protein MCARE in fast-twitch muscle. The first half of Results show that MCARE binds to and regulate activity of SERCA1 and thereby muscle relaxation. In the second half, the authors described that MCARE knockout caused muscle atrophy, reduced muscle strength, elevated serum creatine kinase (CK) and exacerbated exercise-induced muscle damage. However, the studies using MCARE knockout mice were preliminary, lacking mechanistic analysis linking the muscle defects to the molecular role of MCARE.

I have the following concerns:

1. The subcellular location was determined using tagged overexpression. As overexpression of may sometime lead to accumulation of unsorted proteins in ER, where MCARE presumably resides, it is necessary to check the cellular location of natively expressed MCARE.

2. In Figs. 2 and 4a-c, the Ca transient amplitude is only around 0.5 $\Delta F/F_0$ in control or even 0.2 in the MCARE group. Several details need to be clarified for the reliability of these measurements: 1) It is important to determine whether MCARE influence the background intracellular Ca level. This reviewer noticed that in the fluorescence video the MCARE group exhibited spontaneous activity and higher fluorescence around nucleus-like structures, which may elevate the average resting Ca level (F_0). 2) What percentage of cells responded to EPS? In the displacement video, only a small portion of myotubes exhibited apparent displacement. Did you include the "no response" myotubes in the statistical data? 3) Did you record Ca and displacement simultaneously? What $\Delta F/F_0$ values did the peak reach in the myotubes that displayed apparent displacements? 4) What was the composition of the extracellular solution in which the Ca/displacement were measured?

3. Figs. 2g-h reanalyzed the contraction data using analysis of covariance, and compared the velocity/displacement ratio. What's the physical/physiological meaning of the velocity/displacement ratio? How to interpret the real change occurring behind this ratio? To this reviewer, this ratio is simply a parameter of time. Therefore, it is more realistic and much easier for understanding to measure time parameters directly, e.g., by fitting the time constant of relaxation.

4. Why were relaxation velocity and displacement not measured in MCARE knockout mice? Because MCARE regulation of SERCA1 and muscle relaxation is a main point of this study, it is necessary to confirm this conclusion in MCARE knockout

mice.

5. As the major problem of this manuscript, the second half of the results (Lines 191-256) were mostly descriptive, describing a set of phenomena without mechanistic link to the main theme of the title. Although the authors tried to attribute some of the morphological and functional changes to the defective Ca homeostasis in Discussion, these are mostly speculations.

Version 1:

Reviewer comments:

Reviewer #1

(Remarks to the Author)

The manuscript is acceptable.

Reviewer #2

(Remarks to the Author)

The main focus of this manuscript is a novel SR membrane protein that regulates SR Ca²⁺ content and, consequently, excitation-contraction (EC) coupling. The use of knockout (KO) mice provides strong support for the biochemical findings and significantly strengthens the overall study. While the new data improve the manuscript and therefore is acceptable for publication, the continued reliance on C2C12 cells for calcium studies remains a concern. This approach is a notable limitation, as it detracts from the impact of the findings and raises questions about their physiological relevance. The authors justify the lack of studies in isolated muscle fibers by citing damage in KO mice. However, by the same logic, manipulating MCARE in C2C12 cells should also be approached with caution. Demonstrating that the effects observed in C2C12 myotubes are recapitulated in isolated muscle fibers would greatly strengthen the proposed model. Electroporation of the identical constructs would offer such opportunity. It is important to note that C2C12 cells lack the complex architecture of the sarcoplasmic reticulum (SR) and neuromuscular junction (NMJ) found in mature muscle fibers. At a minimum, the authors should acknowledge this limitation in their calcium imaging studies.

Reviewer #3

(Remarks to the Author)

Most of my questions have been answered, and the overall quality of the manuscript has been improved after revision. I have the following remaining concerns:

Q2-2: The authors said that "only EPS-responsive cells were included in the statistical analyses." This is not appropriate, because the ratio of no-response cells also matters.

Q2-4: The authors said that "For calcium imaging, myotubes were incubated and stimulated in FluoroBrite DMEM (Thermo Fisher Scientific) supplemented with 10% FBS and 4 mM L-glutamine". Is the buffer sensitive to CO₂? How the pH was maintained?

Q5: 5. The problem that the second half of the results were mostly descriptive have not been solved.

Version 2:

Reviewer comments:

Reviewer #2

(Remarks to the Author)

The authors have been responsive. This is relevant work for muscle physiologists.

Reviewer #3

(Remarks to the Author)

No further comments.

Draft Only

General response:

We thank the Reviewers for their thoughtful evaluation of our manuscript and for the constructive comments and suggestions, which helped us to improve the clarity and quality of the work.

We have carefully addressed all points raised and revised the manuscript accordingly. In addition to addressing all reviewer comments, we have made several minor but important editorial changes throughout the manuscript to improve clarity and ensure accurate scientific representation. First, we revised the manuscript title from “MCARE is a SERCA activator in fast-twitch skeletal muscle” to “MCARE enhances SERCA1 activity in fast-twitch muscle to maintain calcium handling and muscle integrity.” This revision was made to avoid the implication that MCARE directly activates SERCA1, and to more accurately reflect our mechanistic findings that MCARE enhances SERCA1 activity by displacing MRLN. The revised title also emphasizes the specific role of MCARE in skeletal muscle. We also refined some terminology for clarity and consistency—for example, replacing “fragility” with “vulnerability” when describing the mechanical stress susceptibility of skeletal muscle. Furthermore, we adjusted the order and phrasing of certain explanatory sections to improve logical flow and readability. These changes do not affect the interpretation of our data or the study’s overall conclusions but are intended to enhance the precision and accessibility of the manuscript.

Below, we provide point-by-point responses to the comments from each reviewer.

Reviewer #1 (Remarks to the Author):

In the manuscript by Sasaki et al. entitled "MCARE is a SERCA activator in fast-twitch skeletal muscle", the authors identified the role of the protein MCARE in skeletal muscle. By using in vitro experiments and generating knockout mice, they showed that MCARE is specifically expressed in fast skeletal muscle and interacts with Serca1 to activate the reuptake of cytoplasmic calcium into the sarcoplasmic reticulum.

Overall, this is a comprehensive study with convincing results, the design of the experiments is good, and the conclusions are solid. The article is well written, and I favor the publication of this article in the Nature Communication journal without significant revision.

Author response:

We sincerely appreciate the reviewer’s thoughtful and positive evaluation of our study. We are

grateful for their recognition of the comprehensiveness of our work, the robustness of our results, and the clarity of our manuscript. We are particularly pleased that the reviewer found our experimental design to be well-constructed and our conclusions to be solidly supported by the data. Furthermore, we deeply appreciate their endorsement for publication in Nature Communications without significant revision, which encourages us greatly.

However, there are minor suggestions that could improve the article:

-In line 120, the authors could add a simple scheme in supplemental data to show the model of MCARE structure in the SR membrane.

Author response:

We appreciate the reviewer's suggestion and have included a schematic model of MCARE in the SR membrane in the supplemental data (Extended Data Fig. 2d). We believe this addition improves the clarity of our study and thank the reviewer for their constructive feedback.

-In line 182, the authors could again add a scheme in supplemental data showing the MCARE gene (with exon and intron) and the localization of the sgRNA used to generate the MCARE knock out mouse.

Author response:

We thank the reviewer's suggestion. To address this, we have included a new schematic figure in the new extended data Fig.7a that depicts the MCARE gene structure (exons and introns) and indicates the position of the sgRNA used for MCARE knockout mouse generation.

-In line 191, this phenotype could be better highlighted. Could these data have been presented in the main figure and not just as a supplementary video? For example, pictures of WT and KO muscle over time after dissection could be presented. Or measuring the continued contraction duration.

Author response:

We sincerely appreciate the reviewer's suggestion to better highlight the rippling muscle contraction phenotype. While we attempted to illustrate the muscle movement using static

images, we found that they were insufficient to fully capture the dynamic nature of the phenomenon. As an alternative, we have now incorporated quantitative data on the duration of spontaneous muscle movement in the new Fig. 5h, as recommended. Additionally, we have revised the Results section to include further details of this observation.

Reviewer #2 (Remarks to the Author):

Ca²⁺ signaling has long been recognized as a regulator of muscle contraction through the process of excitation contraction coupling (ECC). ECC involves the dynamic release and uptake of Ca²⁺ from the sarcoplasmic reticulum (SR). Contraction results when Ca²⁺ is released through the RYR channels located in the SR membrane and then Ca²⁺ binds to the myofibril apparatus to stimulate action-myosin ATPase activity. Relaxation involves re-sequestration of Ca²⁺ from the myofibril to the SR by the SERCA pump. Details of the SERCA activity is therefore of great interest in understanding muscle physiology and contractility.

In the current work, the authors have identified a novel transmembrane protein TMEM233 as a regulator of SERCA function that has been renamed MCARE. MCARE therefore joins a growing number of micropeptides that regulate SERCA function. Myoregulin, sarcolipin and phospholamban act as endogenous inhibitors of SERCAs. Their different patterns of tissue expression for these micropeptides confers differential regulation of the specific SERCA. In contrast DWORF, Neuronatin and STIM1 have been shown to restrict access of these micropeptides to SERCA and thereby modulates SERCA activity. In the current work the authors propose that MCARE acts in a similar manner to prevent MLN access to the SERCA1 in fast twitch muscles. The work is significant, if validated, and offers new and important information of Ca²⁺ signaling in muscle.

Author response:

We sincerely appreciate the reviewer's insightful comments on the significance of our study and the broader context of SERCA regulation in muscle physiology. As noted, the regulation of SERCA activity is a critical determinant of Ca²⁺ homeostasis and muscle contractility. In this work, we identify MCARE as a novel regulator of SERCA1 in fast-twitch muscles and demonstrate its role in promoting Ca²⁺ reuptake into the sarcoplasmic reticulum. Our findings suggest that MCARE enhances SERCA1 function by displacing Myoregulin, thereby relieving

its inhibitory effect. We believe it provides new insights into the modulation of Ca²⁺ dynamics in skeletal muscle. We appreciate the reviewer's recognition of the significance of this work and its potential contributions to the field.

1) Prior studies of TMEM233 identified it as an interacting partner for Nav1.7 where it can modulate gating properties of sodium channels. It has also been shown to be the target of the toxin ExTxA. Given the key role of Na channels in muscle excitability (particularly for EPS) how do the authors reconcile their data with the recent publication by Jami et al (PMID: 37117223)? This seems like a key issue. The authors stimulate cells with electrical pulses (EPS). It is possible that MCARE and Nav channels interaction may explain results of altered Ca²⁺ transients. Can the authors utilize the toxin to test their hypothesis. Does blocking Nav channels in muscle influence MCARE actions?

Author response:

We acknowledge the reviewer's insightful comment regarding the potential interaction between MCARE (TMEM233) and Nav channels, as described by Jami et al. (PMID: 37117223). Unfortunately, ExTxA is not commercially available, and we are unable to obtain it for experimental validation.

In skeletal muscle, MCARE is localized to the sarcoplasmic reticulum rather than the plasma membrane. Given that Nav channels are primarily expressed on the plasma membrane, particularly concentrated at the neuromuscular junction as recently reported (bioRxiv [Preprint]. 2023 Oct 29;2023.10.24.563837. doi: 10.1101/2023.10.24.563837. PMID: 37961580), a direct interaction between MCARE and Nav channels is unlikely in muscle cells. Furthermore, even if an interaction were to occur, the activation of Nav channels by TMEM233 has been shown to be ExTxA-dependent. Therefore, under our experimental conditions, we consider it unlikely that MCARE/TMEM233 exerts a significant influence on Nav channel activity.

To avoid confusion for readers, we have added the following sentence to the Discussion:

Line 403:

Notably, Tmem233/MCARE exhibits differential intracellular localization and protein interactions in skeletal muscle cells compared to DRG neurons. This suggests that the function of MCARE in skeletal muscle is likely independent of its interaction with Nav channels.

2) The main Ca²⁺ measurements made in this manuscript involve gain and loss of MCARE in

C2C12 cells. These cells are immature muscle, lack motor nerve innervation, and often don't form fully functional SR and T-tubules which may influence the interpretation of the present data. Ideally these experiments are done with isolated fast twitch fibers from mice.

Author response:

We appreciate the reviewer's suggestion to perform Ca^{2+} imaging in isolated fast-twitch fibers. While this approach would be ideal, it is not feasible in MCARE KO mice due to chronic muscle damage, as evidenced by significantly elevated serum CK levels. Given that damaged muscle fibers exhibit altered Ca^{2+} handling and increased leakage, they are not suitable for precise analysis.

In our Ca^{2+} imaging experiments using C2C12 myotubes, we specifically analyzed only those myotubes that exhibited EPS-induced Ca^{2+} transients. This approach allowed us to selectively assess myotubes with functionally developed SR and T-tubule systems, ensuring the robustness of our findings. Furthermore, we have demonstrated that MCARE-dependent alterations in Ca^{2+} transients are also observed in primary myotubes, which undergo more advanced differentiation than C2C12 cells. These results reinforce the physiological relevance of our data despite the inherent limitations of the C2C12 model.

We hope this clarification addresses the reviewer's concerns and further supports the validity of our experimental approach.

Second, more extensive studies would be required to investigate the effects on SERCA1 especially since C2C12 cells express similar levels of both SERCA1 and SERCA2a.

Author response:

We thank the reviewer for raising this important point. To address the concern regarding the relative contributions of SERCA1 and SERCA2, we performed additional co-immunoprecipitation experiments in HEK293 cells, in which Flag-tagged MCARE and myc-tagged SERCA2 were co-expressed. Immunoprecipitation with an anti-Flag antibody did not pull down myc-SERCA2, suggesting that MCARE does not interact with SERCA2 under these experimental conditions (New Extended Data Fig. 6). In contrast, MCARE robustly co-immunoprecipitated with SERCA1 under the same system (Fig. 3a). These findings support the notion that the alterations in calcium signaling observed in C2C12 cells are primarily mediated through the interaction between MCARE and SERCA1.

3) The authors show that overexpressing MCARE in C2C12 cells result in dramatic reduction

in the Ca²⁺ signal (Fig 2 b-C) whereas myotubes from MCARE KO mice have gain in the Ca²⁺ released (Fig 4 b-C). Ca²⁺ transients represent the summation of events that include Ca²⁺ flux across the plasma membrane and Ca²⁺ flux into and out of the SR. Many factors including SERCA1 are involved. The present studies focus on changes to Ca²⁺ release of SR stores, but do not consider the other issues like Ca²⁺ exchange/transport across the PM and Ca²⁺ leak from the SR. It would be better to design protocols that focus on Ca²⁺ uptake. More detailed studies would be needed to demonstrate the specific Ca²⁺ effect. SERCA1 function mainly influences the terminal portion of the Ca²⁺ transient and is usually assessed as the tau. Authors should consider direct SR Ca measurements with GECI like D1ER or Fluo5N. Permeabilized membranes of fibers enable control of Ca²⁺ concentration, ATP availability and SERCA activity.

Author response:

We are grateful for the reviewer's valuable suggestions. We agree that a more detailed dissection of Ca²⁺ handling, including SR Ca²⁺ leak, Ca²⁺ uptake, and membrane transport mechanisms, would further clarify the functional role of MCARE. While we acknowledge the reviewer's recommendation to use permeabilized muscle fibers to directly assess SERCA1 activity under controlled conditions, our primary objective was to evaluate the impact of MCARE on Ca²⁺ transients in response to physiological stimulation. To this end, we chose to use C2C12 myotubes, which are well-suited for electrical pulse stimulation (EPS)-induced Ca²⁺ imaging.

We initially attempted to measure SR Ca²⁺ levels directly using Fluo-5N in C2C12 myotubes, but the fluorescence signal was too weak for reliable quantification under our experimental conditions. Therefore, we shifted to Fluo-8-based cytosolic Ca²⁺ imaging to assess the decay kinetics of Ca²⁺ transients following EPS.

To ensure sufficient signal strength, we applied a high-intensity EPS protocol (23 V, 6 ms), since the standard 2 ms pulse did not produce adequate fluorescence. Notably, this high-intensity stimulation resulted in greater maximal $\Delta F/F_0$ values compared to the standard EPS protocol, which led to a relatively smaller difference in the amplitude of Ca²⁺ transients between the control and MCARE-overexpressing groups (New Extended Data Fig. 3). Using this approach, we found that MCARE-overexpressing cells exhibited a significantly shorter tau, indicating accelerated cytosolic Ca²⁺ clearance (New Fig. 2d). Given that the majority of Ca²⁺ clearance from the cytosol in myotubes is mediated by SERCA, these results support the notion that MCARE enhances SERCA activity, thereby regulating the decay phase of the Ca²⁺ transient.

While we acknowledge the value of direct SR Ca²⁺ measurements (e.g., via GECIs such as D1ER or use of permeabilized fibers), we believe that our current approach provides functionally relevant evidence that links MCARE to enhanced Ca²⁺ reuptake via SERCA in an EPS-dependent context.

4) If the MCARE activates SERCA1 function, through binding MRLN then overexpressing MCARE would increase the SR Ca²⁺ stores resulting in greater release following EPS. In contrast, if deletion of MCARE results in constitutive interaction of MRLN and reduce the SR Ca²⁺ stores. The studies in Figure 2 and 4 show the opposite raising concerns about the authors interpretation that MCARE activates SERCA1. It is important to show that in Figures 2 and 4 that MRLN interaction with SERCA1 is affected by the gain and loss of function of MCARE similar to studies in figure 3 (all from 293 cells).

Author response:

We thank the reviewer for raising this important point regarding the apparent discrepancy between MCARE-dependent changes in Ca²⁺ transients and the expected effects on SR Ca²⁺ stores. To further clarify the relationship between SERCA activity and EPS-induced Ca²⁺ dynamics, we performed Ca²⁺ imaging in C2C12 myotubes treated with thapsigargin, a well-characterized SERCA inhibitor (new Extended Data Fig. 5a, b). This treatment resulted in a significant increase in the amplitude of EPS-induced Ca²⁺ transients (Max $\Delta F/F_0$), along with a modest but significant elevation of baseline Ca²⁺ levels (F_0), consistent with impaired reuptake and cytosolic Ca²⁺ accumulation.

These findings indicate that in our experimental system, the Max $\Delta F/F_0$ value reflects not only the amount of Ca²⁺ released from the SR, but also the efficiency of reuptake during EPS stimulation. From this perspective, the reduced Max $\Delta F/F_0$ observed in MCARE-overexpressing myotubes (Fig. 2b, c) can be interpreted as a result of enhanced SERCA function—allowing for faster Ca²⁺ clearance during and between EPS pulses—rather than a reduction in SR Ca²⁺ content. Conversely, the increased Max $\Delta F/F_0$ in MCARE-deficient primary myotubes (Fig. 4b, c) likely reflects impaired SERCA-mediated clearance, leading to a transient accumulation of cytosolic Ca²⁺ during EPS stimulation. Thus, the amplitude of the Ca²⁺ transient is shaped by both SR release and reuptake efficiency, and our findings are consistent with MCARE enhancing SERCA function.

We agree with the reviewer that direct assessment of SR Ca²⁺ content would be ideal to fully disentangle these effects. While we initially attempted SR Ca²⁺ imaging using Fluo-5N in C2C12 myotubes, the signal was too weak for reliable quantification under our conditions. As an alternative, we evaluated the decay kinetics of the Ca²⁺ transients, and found that MCARE overexpression significantly shortened the tau value (new Fig. 2d), supporting an increase in SERCA-dependent Ca²⁺ clearance.

Regarding the MRLN–SERCA1 interaction under MCARE gain- or loss-of-function conditions,

we fully agree that this would be important to demonstrate in muscle cells. However, due to the lack of reliable commercial antibodies that detect endogenous MRLN, we were unable to perform these experiments in C2C12 or native skeletal muscle tissue. Nonetheless, our assays in HEK293 cells (Fig. 3) demonstrate that MCARE competes with MRLN for binding to SERCA1, and together with the Ca²⁺ imaging data described above, support the model in which MCARE enhances SERCA activity through displacement of MRLN.

5) The studies in figure 5 address the response to injury that is associated with exercise. PGC1 and ATF3 levels are increased and exaggerated in the KO mice. How does this relate to the changes in Ca²⁺ flux? Or SERCA1 ATP consumption? Seems like Ca²⁺ levels at the contractile apparatus would not clear efficiently if MCARE activates SERCA1. What happens to calpain activity?

Author response:

We appreciate the reviewer's thoughtful comments regarding the connection between altered Ca²⁺ handling and stress responses following exercise.

Contrary to the reviewer's interpretation, our data show that PGC-1 α expression following exercise is not significantly altered in MCARE KO mice compared to WT controls. As SERCA is known to be one of the most ATP-consuming proteins in skeletal muscle, its reduced activity in the absence of MCARE may lead to lower ATP turnover. However, the lack of change in PGC-1 α expression under both sedentary and exercise conditions suggests that major energy-sensing pathways (e.g., the AMPK–PGC-1 α axis) are not broadly altered in MCARE-deficient muscle.

As indicated by serum CK levels in WT and KO mice, skeletal muscles of MCARE KO mice exhibit mild but chronic muscle damage, which is further exacerbated by treadmill exercise. Since ATF3 is known to be upregulated in response to muscle injury, the increased ATF3 expression observed in KO mice likely reflects the extent of muscle damage. Although ATF3 can also be induced by various stressors such as oxidative stress and ER stress, we found no substantial differences in the expression

Rebuttal Fig. 1

Expression levels of *HO-1*, *BiP* and *Chop* mRNA in TA muscles of WT and KO mice, both at baseline and after 30 min of treadmill running. n = 6.

of marker genes for these stress pathways between WT and KO mice (Rebuttal Fig. 1). Therefore, the specific upstream signals contributing to the elevated ATF3 expression in KO muscle remain unclear.

We agree with the reviewer that calpain activity could potentially be elevated in MCARE KO mice due to impaired Ca²⁺ clearance and may contribute to muscle injury. However, as noted in previous review articles (e.g., the section on “Measurements of calpain activity” in *Int J Sports Med.* 2020 Dec;41(14):994-1008.), accurate assessment of in vivo calpain activity remains technically challenging. In particular, Ca²⁺ release from the sarcoplasmic reticulum during tissue homogenization can artificially activate calpains, complicating interpretation. Further studies using more refined methods will be required to determine whether calpains are indeed contributing to muscle damage in the absence of MCARE.

Line: 338

In response to this comment, we have added the following sentence to the Discussion: “Given its role in exercise-induced muscle damage, calpain—a Ca²⁺-activated protease—may also contribute to the pathological features observed in MCARE-deficient muscle²³. Future studies should explore whether impaired Ca²⁺ homeostasis leads to aberrant calpain activation in this context.”

Reviewer #3 (Remarks to the Author):

This manuscript identified transmembrane protein MCARE in fast-twitch muscle. The first half of Results show that MCARE binds to and regulate activity of SERCA1 and thereby muscle relaxation. In the second half, the authors described that MCARE knockout caused muscle atrophy, reduced muscle strength, elevated serum creatine kinase (CK) and exacerbated exercise-induced muscle damage. However, the studies using MCARE knockout mice were preliminary, lacking mechanistic analysis linking the muscle defects to the molecular role to MCARE.

Author response:

We thank Reviewer #3 for their thoughtful and constructive feedback. We sincerely appreciate their recognition of our identification of MCARE as a transmembrane protein that regulates SERCA1 activity and contributes to skeletal muscle function. We also acknowledge and value

the reviewer's concern regarding the need for a clearer mechanistic link between the molecular role of MCARE and the muscle phenotypes observed in knockout mice. In response, we have revised the manuscript to better clarify this connection and have addressed the reviewer's comments point-by-point below.

I have the following concerns:

1. The subcellular location was determined using tagged overexpression. As overexpression of may sometime lead to accumulation of unsorted proteins in ER, where MCARE presumably resides, it is necessary to check the cellular location of natively expressed MCARE.

Author response:

We appreciate the reviewer's insightful comment regarding the potential artifact of subcellular localization analysis based on overexpression. We fully agree that overexpression may sometimes lead to non-physiological localization, particularly to the ER/SR.

To examine the localization of endogenously expressed MCARE, we initially attempted immunofluorescent staining using skeletal muscle cryosections. However, we found that the MCARE antibody generated in our laboratory was not suitable for specific detection of the MCARE protein by immunostaining (Data not shown), and we therefore abandoned this approach.

To overcome this limitation, we prepared microsomal fractions from skeletal muscle tissue using differential centrifugation. We successfully enriched the sarcoplasmic reticulum (SR) compartment, as evidenced by the presence of SERCA1, and found that MCARE was also highly enriched in the same fraction (New Fig. 1f). Minimal contamination from other organelle markers supports the specificity of this fractionation. These results indicate that endogenously expressed MCARE localizes predominantly to the SR membrane.

2. In Figs. 2 and 4a-c, the Ca transient amplitude is only around 0.5 $\Delta F/F_0$ in control or even 0.2 in the MCARE group. Several details need to be clarified for the reliability of these measurements: 1) It is important to determine whether MCARE influence the background intracellular Ca level. This reviewer noticed that in the fluorescence video the MCARE group exhibited spontaneous activity and higher fluorescence around nucleus-like structures, which may elevate the average resting Ca level (F_0).

Author response:

We appreciate the reviewer's insightful comment regarding the potential influence of MCARE on baseline intracellular Ca^{2+} levels. We also noticed stronger fluorescence signals near perinuclear regions in MCARE-overexpressing C2C12 myotubes. We speculate that this may result from enhanced SR Ca^{2+} uptake due to MCARE-mediated SERCA activation, leading to a localized cycle of Ca^{2+} leak and reuptake that elevates perinuclear Ca^{2+} concentration.

As the reviewer correctly points out, accurate interpretation of $\Delta F/F_0$ requires careful consideration of resting fluorescence (F_0). In response, we have now included the F_0 values in the relevant figures to complement the $\Delta F/F_0$ analysis. In the revised Fig. 2b and new Extended Data Fig. 3b, we show that although strong perinuclear fluorescence was occasionally observed, the average F_0 values across the entire myotube did not differ significantly between MCARE-overexpressing and control groups.

In contrast, Extended Data Fig. 5a shows a modest but statistically significant increase in F_0 following thapsigargin treatment. Despite this increase, the maximum $\Delta F/F_0$ was also significantly higher, indicating that impaired SERCA function enhances the amplitude of EPS-induced Ca^{2+} transients.

Similarly, primary myotubes from *Mcare* KO mice exhibited significantly elevated F_0 (Fig. 4b), yet the maximum $\Delta F/F_0$ was also increased. These results support the conclusion that MCARE deficiency impairs Ca^{2+} reuptake, thereby contributing to a transient increase in cytosolic Ca^{2+} amplitude during EPS stimulation, rather than merely reflecting elevated baseline fluorescence.

2) What percentage of cells responded to EPS? In the displacement video, only a small portion of myotubes excited apparent displacement. Did you include the "no response" myotubes in the statistical data?

Author response:

We thank the reviewer for raising this important point. It is well recognized that C2C12 myotubes differentiated under standard conditions do not uniformly develop contractile responsiveness, and prolonged EPS stimulation (several hours to overnight) is often required to induce visible contractions. Even under optimized conditions, only approximately 10–30% of myotubes exhibit clear displacement in response to EPS.

To ensure accurate evaluation of MCARE's effects in functionally contractile myotubes, only EPS-responsive cells were included in the statistical analyses. We believe this approach allows for a more precise assessment of MCARE's physiological role.

To clarify this point for readers, we have added the following sentence to the Methods section:

(Line: 784)

“Myotubes that did not exhibit any observable contraction in response to EPS were considered non-responsive and were excluded from the analysis.”

3) Did you recorded Ca and displacement simultaneously? What $\Delta F/F_0$ values did the peak reach in the myotubes that displayed apparent displacements?

Author response:

We appreciate the reviewer’s question regarding simultaneous measurement. During the course of this study, we observed that while many C2C12 myotubes displayed EPS-induced calcium transients under standard differentiation conditions, acquisition of contractile ability required additional, prolonged EPS exposure. Therefore, the calcium imaging and motion analysis experiments were conducted using different populations of C2C12 myotubes subjected to different protocols.

As a result, we did not simultaneously record calcium transients and displacement in the same myotubes. However, we fully agree that correlating intracellular calcium dynamics with mechanical output at the single-cell level represents an important future direction, and we intend to pursue this line of investigation in subsequent work.

4) What was the composition of the extracellular solution in which the Ca/displacement were measured?

Author response:

We thank the reviewer for the question.

For calcium imaging, myotubes were incubated and stimulated in FluoroBrite DMEM (Thermo Fisher Scientific) supplemented with 10% FBS and 4 mM L-glutamine, which was selected based on the manufacturer’s recommendation for optimal imaging conditions under serum-containing media. For motion analysis, myotubes were maintained and stimulated in DMEM supplemented with 2% horse serum. These experimental conditions are now clearly described in the revised Methods section.

3. Figs. 2g-h reanalyzed the contraction data using analysis of covariance, and compared the velocity/displacement ratio. What’s the physical/physiological meaning of the

velocity/displacement ratio? How to interpret the real change occurring behind this ratio? To this reviewer, this ratio is simply a parameter of time. Therefore, it is more realistic and much easier for understanding to measure time parameters directly, e.g., by fitting the time constant of relaxation.

Author response:

We thank the reviewer for this insightful comment.

In the course of analyzing muscle contraction and relaxation kinetics, we observed a strong correlation between contraction/relaxation velocity and displacement. Based on this, we considered displacement to be a covariate that could confound the interpretation of velocity differences between groups. Therefore, we performed analysis of covariance (ANCOVA) to compare contraction and relaxation velocities while statistically controlling for displacement. The velocity/displacement ratio was presented as a normalized indicator of contraction or relaxation speed independent of displacement amplitude. We believe this approach provides a mathematically rigorous assessment of how MCARE affects relaxation kinetics.

However, we fully agree with the reviewer that the physiological and physical interpretation of this ratio is not straightforward, and may not be intuitive for all readers. In light of this, and following the reviewer's helpful suggestion, we have included a complementary analysis using direct time-based parameters.

Specifically, we generated plots showing the decline in contraction over time following the peak of contraction, with the vertical axis representing the percentage of peak contraction and the horizontal axis representing time. From these data, we calculated the half-relaxation time (T_{50}), defined as the time required for the contraction to decay to 50% of its maximum value. These results are now shown in Fig. 2j and Fig. 4h.

The inclusion of T_{50} provides a more physiologically interpretable parameter and further supports our conclusion that MCARE is a key regulator of muscle relaxation kinetics.

4. Why were relaxation velocity and displacement not measured in MCARE knockout mice? Because MCARE regulation of SERCA1 and muscle relaxation is a main point of this study, it is necessary to confirm this conclusion in MCARE knockout mice.

Author response:

We thank the reviewer for this important comment and fully agree on the necessity of validating MCARE's role in muscle relaxation using *Mcare* KO mice. In response, we conducted motion analysis of primary myotubes isolated from WT and MCARE KO mice following EPS

stimulation

While maximum contraction and relaxation velocities and contraction displacement did not significantly differ between WT and KO groups, more detailed analyses revealed meaningful differences. Specifically, ANCOVA using contraction displacement as a covariate, as well as normalization of velocities by displacement, showed a significant reduction in relaxation velocity in *Mcare* KO-derived myotubes (Fig. 4d–g).

Furthermore, we performed time-based analysis by plotting the decline in contraction over time and calculating the half-relaxation time (T_{50}). This analysis revealed a significantly prolonged T_{50} in KO myotubes compared to WT (Fig. 4h), indicating delayed relaxation.

Together, these results provide strong evidence that endogenous MCARE is essential for maintaining normal muscle relaxation kinetics.

5. As the major problem of this manuscript, the second half of the results (Lines 191-256) were mostly descriptive, describing a set of phenomena without mechanistic link to the main theme of the title. Although the authors tried to attribute some of the morphological and functional changes to the defective Ca homeostasis in Discussion, these are mostly speculations.

Author response:

We appreciate the reviewer's critical and constructive feedback regarding the mechanistic interpretation of the phenotypes observed in *Mcare* KO mice. While the first half of our study focuses on defining the molecular function of MCARE in enhancing SERCA activity using in vitro models, the second half aims to establish the physiological relevance of this regulation in vivo.

Given the essential role of SERCA in Ca^{2+} homeostasis and the broad impact of Ca^{2+} signaling on skeletal muscle function, we believe that the phenotypes observed in *Mcare*-deficient mice—such as progressive muscle atrophy, rippling contraction, and increased muscle vulnerability—are consistent with impaired SERCA function and support a mechanistic link.

That said, we fully agree with the reviewer that definitive, causal evidence directly connecting reduced SERCA activity to these phenotypes remains lacking. We have revised the Discussion section to explicitly acknowledge this limitation and to emphasize the need for future studies to dissect the downstream mechanisms involved.

Line 352:

“Although our findings provide important insights into the physiological role of MCARE in skeletal muscle, we acknowledge that the precise molecular mechanisms linking reduced SERCA activity to the diverse phenotypes observed in *Mcare* KO mice remain to be fully elucidated.

Further investigation will be required to delineate the downstream pathways that connect impaired Ca²⁺ handling to muscle atrophy, increased vulnerability, and rippling contractions.”

Draft Only

We are grateful to the reviewers for the careful and thoughtful assessment of our revised manuscript.

We sincerely appreciate their time, efforts, and valuable feedback, which have significantly contributed to strengthening our work.

Below, we provide point-by-point responses to the reviewers' comments, with corresponding changes indicated in the revised manuscript using tracked changes.

Reviewer #1 (Remarks to the Author):

The manuscript is acceptable.

Author response:

We thank the reviewer for their continued support and for finding the revised version of our manuscript acceptable.

Reviewer #2 (Remarks to the Author):

The main focus of this manuscript is a novel SR membrane protein that regulates SR Ca²⁺ content and, consequently, excitation-contraction (EC) coupling. The use of knockout (KO) mice provides strong support for the biochemical findings and significantly strengthens the overall study. While the new data improve the manuscript and therefore is acceptable for publication, the continued reliance on C2C12 cells for calcium studies remains a concern. This approach is a notable limitation, as it detracts from the impact of the findings and raises questions about their physiological relevance. The authors justify the lack of studies in isolated muscle fibers by citing damage in KO mice. However, by the same logic, manipulating MCARE in C2C12 cells should also be approached with caution. Demonstrating that the effects observed in C2C12 myotubes are recapitulated in isolated muscle fibers would greatly strengthen the proposed model. Electroporation of the identical constructs would offer such opportunity. It is important to note that C2C12 cells lack the complex architecture of the sarcoplasmic reticulum (SR) and neuromuscular junction (NMJ) found in mature muscle fibers. At a minimum, the authors should acknowledge this limitation in their calcium imaging studies.

Author response:

We thank the reviewer for the valuable suggestion to validate our findings using isolated muscle fibers. In response, we conducted a preliminary calcium imaging experiment using isolated muscle fibers from wild-type mice loaded with the fluorescent indicator Fluo-8 AM. However,

the fluorescence signal was extremely weak or highly uneven, with strong signals observed only in restricted regions within the fibers (Rebuttal Fig. 1). This heterogeneity is likely due to technical limitations, such as variability in dye uptake or intracellular esterase activity. Therefore, under our current experimental conditions, we judged that consistent and reliable measurements of cytosolic Ca^{2+} in isolated muscle fibers are difficult to achieve. We therefore hope for the reviewer's understanding regarding our reliance on C2C12 myotubes for calcium imaging in this study.

In addition, we have revised the manuscript to acknowledge this limitation. Specifically, we added the following sentence to the Discussion section to address the concern

Revised text (Line 383):

“In addition, we recognize that our calcium imaging data, which were primarily obtained using C2C12 myotubes, are subject to limitations due to the absence of complex SR and neuromuscular junction architecture in these cells.”

Reviewer #3 (Remarks to the Author):

Most of my questions have been answered, and the overall quality of the manuscript has been improved after revision. I have the following remaining concerns:

Author response:

We sincerely thank the reviewer for their careful reading of our revised manuscript and for recognizing the overall improvement in its quality. We appreciate the reviewer's constructive comments throughout the review process, which have been invaluable in strengthening the manuscript. Below, we provide point-by-point responses to the remaining concerns raised.

Rebuttal Fig. 1

Isolated skeletal muscle fibers were loaded with the calcium-sensitive dye Fluo-8 AM and stimulated by electrical pulse stimulation (EPS). Fluorescence images show the distribution of cytosolic Ca^{2+} signals during EPS. Fluorescence signals appeared weak or spatially heterogeneous, highlighting the technical limitations of calcium imaging in isolated muscle fibers under these conditions.

Q2-2: The authors said that “only EPS-responsive cells were included in the statistical analyses.” This is not appropriate, because the ratio of no-response cells also matter.

Author response:

We thank the reviewer for pointing out this important issue. In response, we quantified the proportion of EPS-responsive cells in both C2C12 myotubes (Fig. 2b) and primary myotubes derived from WT and KO mice (Fig. 4b). These analyses revealed no significant differences in the proportion of responsive myotubes between the respective groups. We have added these data to the revised manuscript to address the reviewer’s concern that the ratio of non-responsive cells also matters.

Q2-4: The authors said that “For calcium imaging, myotubes were incubated and stimulated in FluoroBrite DMEM (Thermo Fisher Scientific) supplemented with 10% FBS and 4 mM L-glutamine”. Is the buffer sensitive to CO₂? How the pH was maintained?

Author response:

We apologize for the lack of clarity in our initial description. FluoroBrite DMEM is a CO₂-dependent medium containing sodium bicarbonate as a buffering component. To minimize pH fluctuations during imaging under CO₂-independent conditions, the medium was pre-equilibrated in a 5% CO₂ incubator at 37 °C for at least 30 minutes before use, and after medium replacement the cells were further incubated for 5 minutes to stabilize pH and temperature. All imaging procedures were then completed within 5 minutes after removing the plates from the incubator. This approach ensured that pH changes did not significantly affect our fluorescence measurements.

To clarify this point, we have revised the Methods section as follows:

Revised text (Line 831):

“The medium was pre-equilibrated in a 5% CO₂ incubator at 37 °C for at least 30 min before use, and after replacement the cells were further incubated for 5 min to stabilize pH and temperature prior to imaging. The cells were then stimulated with electric pulses of 23 V for 2 ms or 6 ms. The intracellular calcium responses were monitored using an all-in-one fluorescence microscope (BZ-X810 system, Keyence). All imaging procedures were completed within 5 min after removing the plate from the incubator, which minimizes pH fluctuations under CO₂-independent conditions.”

Q5: 5. The problem that the second half of the results were mostly descriptive have not been solved.

Author response:

We thank the reviewer for raising this critical point once again. We fully acknowledge that the second half of our study remains largely phenomenological in nature and does not provide definitive mechanistic evidence directly linking MCARE loss to the observed pathological features in vivo. This limitation reflects the complexity of in vivo skeletal muscle physiology and the current technical challenges in selectively restoring or modulating SERCA activity in MCARE-deficient muscle in a temporally controlled manner. Indeed, even in Brody disease, a human myopathy caused by mutations in SERCA1, patients exhibit delayed muscle relaxation along with reduced muscle mass, muscle weakness, and elevated serum CK levels. However, the precise molecular mechanisms underlying these diverse symptoms remain poorly understood. This clinical precedent underscores both the challenges of definitively linking SERCA dysfunction to specific pathological features in vivo and the plausibility of our working model that SERCA dysregulation can result in a constellation of muscular pathologies.

While we were unable to definitively delineate the mechanistic pathways linking MCARE loss to the observed phenotypes, our study nevertheless provides several important insights. Notably, we found no upregulation of classical atrophy-related genes such as Atrogin-1 and MuRF1 in *Mcare* KO mice, suggesting that the observed muscle atrophy is mechanistically distinct from that induced by disuse, denervation, or systemic catabolic states. Furthermore, histological and functional assessments revealed no abnormalities in muscle regeneration, indicating that regenerative capacity is preserved despite chronic muscle pathology. In addition, as shown in the newly added extended data, we determined that the exacerbation of exercise-induced muscle damage in KO mice is unlikely to be mediated by classical oxidative or ER stress pathways, thereby narrowing the potential mechanisms underlying this vulnerability and providing a more focused framework for future mechanistic exploration.

To better address the reviewer's concern that the latter part of our study remained largely descriptive, we made the following five substantive revisions aimed at enhancing the mechanistic context and interpretive depth of our findings:

1. Clarifying mechanistic context at the start of the phenotype section:

We have clarified the mechanistic context at the beginning of the phenotype section to better connect impaired Ca^{2+} handling with the observed in vivo phenotypes. Specifically, we now

introduce the phenotypic characterization section of the Results by highlighting that mutations in SERCA1 cause Brody disease, a human myopathy characterized by delayed muscle relaxation, muscle atrophy, weakness, and elevated serum CK levels. This addition provides clinical precedence for the idea that SERCA dysfunction can result in a constellation of muscular pathologies—not limited to contractile defects—and strengthens the plausibility of our working model.

Revised text (Line 237):

“Given that reduced SERCA activity is suggested in the skeletal muscle of *Mcare* KO mice (Fig. 4a-c), we next investigated whether impaired Ca^{2+} reuptake contributes to muscle pathology in vivo. Notably, mutations in SERCA1 cause Brody disease, a human myopathy characterized not only by delayed muscle relaxation but also by reduced muscle mass and strength, along with elevated serum CK levels^{14, 15}. These clinical features suggest that impaired SERCA1 function can lead to complex muscle dysfunction beyond simple contractile defects. *Mcare*-deficient mice showed similar food intake, energy expenditure, and RER compared to WT mice (Extended Data Fig. 8a, b), and there were no significant differences in body weight, liver weight, or white adipose tissue weight between the two groups (Extended Data Fig. 9). Despite these similarities, KO mice exhibited progressive muscle atrophy, especially in proximal muscles such as the quadriceps and gastrocnemius, whereas the TA muscle, which also expresses high levels of MCARE, did not show significant atrophy (Fig. 5a). Notably, significant weight changes in the quadriceps were evident as early as three months of age.”

We believe this revision improves the interpretive framework of the Results section and better supports our hypothesis that MCARE-mediated regulation of SERCA1 is essential for maintaining muscle structure and function.

2. Incorporating new analyses of *Atf3* regulation:

To refine our interpretation of the MCARE-dependent increase in *Atf3* expression after exercise, we measured mRNA levels of *Heme oxygenase-1 (HO-1)*, *binding immunoglobulin protein (BiP)*, and *C/EBP homologous protein (Chop)*, canonical markers of oxidative and ER stress. None of these genes exhibited expression changes that paralleled *Atf3* induction, suggesting that these stress pathways are unlikely to account for the differential *Atf3* response in KO mice (Extended Data Fig. 13). This new analysis excludes major classical stress pathways and points toward an alternative mechanism—potentially a heightened physiological burden on MCARE-deficient muscle—as the basis for the observed *Atf3* response.

Revised text (Line 306):

“To determine whether classical stress pathways contribute to the MCARE-dependent increase in Atf3 expression, we measured the mRNA levels of Heme oxygenase-1 (HO-1), binding immunoglobulin protein (BiP), and C/EBP homologous protein (Chop), which are canonical markers of oxidative and ER stress. None of these genes exhibited expression changes consistent with the Atf3 induction, indicating that these classical stress pathways are unlikely to explain the differential Atf3 response observed in KO mice (Extended Data Fig. 13). This exclusion narrows the possible mechanisms and suggests that the greater Atf3 induction in KO mice more likely reflects an increased physiological burden on fast-twitch skeletal muscle fibers lacking MCARE, rather than activation of classical oxidative or ER stress responses.”

3. Rewriting the Discussion to clarify the speculative nature of our mechanistic interpretations:

In the Discussion section, we now more clearly articulate the speculative nature of the mechanistic links proposed between impaired Ca²⁺ homeostasis and muscle atrophy or rippling contractions, and explicitly position these interpretations as hypotheses for future testing.

Revised text (Line 372):

“Although our findings provide important insights into the physiological role of MCARE in skeletal muscle, we acknowledge that the precise molecular mechanisms linking reduced SERCA1 activity to the diverse phenotypes observed in *Mcare* KO mice remain to be fully elucidated. In particular, the connections between impaired Ca²⁺ homeostasis and downstream events such as muscle atrophy, increased susceptibility to damage, and rippling contractions are currently speculative. These observations raise several hypotheses regarding the involvement of calcium-sensitive proteases (e.g., calpains), altered excitation-contraction coupling, or stress-responsive signaling pathways, all of which warrant future investigation using temporally and spatially controlled genetic or pharmacological approaches. Further studies will be necessary to dissect these downstream pathways and determine whether they causally link MCARE deficiency to the observed pathological features.”

4. Revising the subheading:

We have revised the subheading from “*Mcare* deficiency causes muscle dysfunction in mice” to “*Mcare* deficiency impairs Ca²⁺ handling and induces muscle dysfunction in vivo” to more accurately reflect the core findings presented in this section.

This change highlights the mechanistic insight provided by our calcium imaging and motion analysis data, which show that MCARE deficiency leads to dysregulated intracellular Ca²⁺ handling—presumably due to impaired SERCA1 function—and contributes to subsequent muscle dysfunction.

We believe this revised subheading more clearly communicates the biological relevance of our findings and aligns with the reviewer’s request to emphasize potential mechanistic links between SERCA dysfunction and the observed phenotypes.

5. Restructuring the Results section:

To improve clarity and thematic consistency, we have restructured the Results section to better highlight the potential relationships between defective SERCA function and the observed phenotypes in *Mcare* KO mice, even if causality cannot yet be fully established.

Revised text (Line 250):

“The absence of classical atrophy-related gene upregulation suggests that this phenotype is mechanistically distinct from that induced by fasting, denervation, or inactivity, and may reflect chronic perturbations in Ca²⁺ homeostasis due to impaired SERCA1 regulation (Extended Data Fig. 10).”

Revised text (Line 323):

“These data indicate that MCARE had little effect on muscle regeneration after injury, highlighting that the muscle pathology observed in KO mice is likely caused by Ca²⁺ handling abnormalities through mechanisms independent of, rather than due to, defective regeneration. ”

We hope that these revisions help better contextualize the descriptive aspects of our in vivo findings and strengthen the mechanistic coherence of the manuscript as a whole.